# A lncRNA signature associated with tumor immune heterogeneity predicts distant metastasis in locoregionally advanced nasopharyngeal carcinoma

Ye-Lin Liang[1,2,6], Yuan Zhang[1,2,6], Xi-Rong Tan[1,3,6], Han Qiao[1,3,6], Song-Ran Liu[1,4,6], Ling-Long Tang[1,2], Yan-Ping Mao [1,2], Lei Chen[1,2], Wen-Fei Li[1,2], Guan-Qun Zhou[1,2], Yin Zhao [1,3], Jun-Yan Li[1,2], Qian Li[1,2], Sheng-Yan Huang[1,3], Sha Gong[1,3], Zi-Qi Zheng[1,2], Zhi-Xuan Li[1,2], Ying Sun [1,2], Wei Jiang [5✉], Jun Ma [1,2✉], Ying-Qin Li [1,3✉] & Na Liu [1,3✉]

Increasing evidence has revealed the roles of long noncoding RNAs (lncRNAs) as tumor biomarkers. Here, we introduce an immune-associated nine-lncRNA signature for predicting distant metastasis in locoregionally advanced nasopharyngeal carcinoma (LA-NPC). The nine lncRNAs are identified through microarray profiling, followed by RT–qPCR validation and selection using a machine learning method in the training cohort ($n = 177$). This nine-lncRNA signature classifies patients into high and low risk groups, which have significantly different distant metastasis-free survival. Validations in the Guangzhou internal ($n = 177$) and Guilin external ($n = 150$) cohorts yield similar results, confirming that the signature is an independent risk factor for distant metastasis and outperforms anatomy-based metrics in identifying patients with high metastatic risk. Integrative analyses show that this nine-lncRNA signature correlates with immune activity and lymphocyte infiltration, which is validated by digital pathology. Our results suggest that the immune-associated nine-lncRNA signature can serve as a promising biomarker for metastasis prediction in LA-NPC.

[1] State Key Laboratory of Oncology in South China, Collaborative Innovation Center of Cancer Medicine, Guangdong Key Laboratory of Nasopharyngeal Carcinoma Diagnosis and Therapy, Sun Yat-sen University Cancer Center, Guangzhou 510060, P.R. China. [2] Department of Radiation Oncology, Sun Yat-sen University Cancer Center, Guangzhou 510060, P.R. China. [3] Department of Experimental Research, Sun Yat-sen University Cancer Center, Guangzhou, China. [4] Department of Pathology, Sun Yat-sen University Cancer Center, Guangzhou, China. [5] Department of Radiation Oncology, Affiliated Hospital of Guilin Medical University, Guilin, China. [6] These authors contributed equally: Ye-Lin Liang, Yuan Zhang, Xi-Rong Tan, Han Qiao, Song-Ran Liu. ✉email: weijiang@glmc.edu.cn; majun2@mail.sysu.edu.cn; liyingq@sysucc.org.cn; liun1@sysucc.org.cn

A pproximately 70% of patients with nasopharyngeal carcinoma (NPC) present with locoregionally advanced disease[1,2]. In recent years, significant improvements in the local control of locoregionally advanced NPC (LA-NPC) have been made with the implication of intensity-modulated radiotherapy (IMRT)[3]. However, 20% of patients develop distant metastasis after radical treatment, making it the major contributor to NPC-associated deaths[4,5]. Currently, the most widely used prognostic indicators for NPC are the tumor-node-metastasis (TNM) stage and circulating Epstein-Barr virus (EBV) DNA load. The TNM stage is an anatomically based system, and circulating EBV DNA consists of short DNA fragments released by NPC cells, the level of which is highly correlated with tumor size. However, it has been reported that patients with the same stage and similar EBV DNA levels experience a variety of clinical outcomes[6], indicating the insufficiency of these anatomical prognostic factors. Moreover, NPC is characterized by abundant infiltration of lymphocytes in tumor[7], indicating the heterogeneous nature and complex regulatory network within the tumor. Therefore, molecular biomarkers may provide more insightful information for NPC prognosis.

The discovery and characterization of long noncoding RNAs (lncRNAs) have gained widespread attention because of their various regulatory functions in biological processes[8]. It is generally recognized that lncRNAs are a group of transcripts that are exquisitely regulated and are more cell-type-specific to a greater degree than mRNA[9]. Accumulating evidence correlates lncRNA dysregulation to human diseases, including cancers[10–12]. To date, a large number of lncRNAs have been reported to facilitate tumor growth[13,14], migration and invasion[15,16], and to modulate immune response[17] and signaling pathways[18], thus contributing to distant metastasis[19]. Consequently, prognostic lncRNA signatures have been developed, and have shown promising accuracy in predicting tumor metastasis[20–22]. However, few studies have yet reported whether lncRNA signatures could serve as metastasis predictors for patients with NPC.

A growing number of studies have confirmed the role of lncRNAs as prognostic biomarkers in multiple cancers, primarily based on RNA-seq and microarray data from public databases[23–26]. However, the biomarkers developed by high-throughput methods lack the generalizability required for clinical translation due to the high cost of RNA-seq and microarray techniques. Quantitative reverse transcription-polymerase chain reaction (RT–qPCR), a universal and cost-effective approach, may make biomarkers feasible for use in most hospitals. However, caution should be taken when using different methods such as RNA-seq and RT–qPCR to generate different formulas and cutoff values due to their technical heterogeneity. Some methods are often difficult to translate to the real world, arguably due to the inaccessibility or complexity of the calculation formula[23,24] and lack of independent cohort validation[25,26]. Hence, an easily measurable prognostic lncRNA biomarker that stands up to validation is highly desirable.

In this multicenter cohort study, we introduce a nine-lncRNA signature as a robust predictor of LA-NPC metastasis. Global lncRNA expression is initially profiled with microarrays in the discovery cohort, which identifies metastasis-related lncRNAs. By applying RT–qPCR assay, we detect metastasis-related lncRNAs in 177 tissue samples from the training cohort and then select nine lncRNAs to construct a signature using a machine learning method. The signature can distinguish patients with increased metastatic risk and provide more accurate prediction than traditional clinical factors, which is validated by two independent cohorts. Bioinformatic analysis is used to explore the immune-associated characteristics of the nine-lncRNA signature and then, digital pathology is applied to confirm that the signature reflects

the immunological heterogeneity within the tumor microenvironment (TME) of NPC. In summary, we identify and validate an immune-associated lncRNA signature that can serve as a promising tool for metastasis prediction in LA-NPC.

## Results

**Patient characteristics**. We included 542 nonmetastatic LA-NPC patients and 18 healthy controls in this study (Fig. 1). In the discovery cohort, 18 LA-NPC patients and 18 healthy controls were matched by age and sex (Supplementary Table 1), and ten pairs of LA-NPC patients developed with or without posttreatment distant metastasis were matched based on additional tumor characteristics (T and N stage) and treatment modalities (Supplementary Table 2). The patient characteristics of the Guangzhou training cohort ($n = 177$), Guangzhou internal validation cohort ($n = 177$), and Guilin external validation cohort ($n = 150$) are shown in Table 1. The median follow-up was 83.5 months (interquartile range (IQR) 61.7–94.9), 82.1 months (IQR 60.3–96.4), and 49.4 months (IQR 39.9–62.0) in the training, internal and external cohorts, respectively.

**Development of a nine-lncRNA signature for NPC metastasis**. Based on the lncRNA microarray data, 1453 lncRNAs were differentially expressed between the tumor and normal tissues, and 525 lncRNAs were differentially expressed between the matched metastatic and nonmetastatic LA-NPC tissues. Overall, there were 149 lncRNAs at the intersection of these two comparisons (Supplementary Data 1), displaying strong classification properties (Supplementary Fig. 1a, b).

To construct a simple and practical prognostic model, we detected the expression of the 149 lncRNAs by RT–qPCR assay and then used a machine learning method to reduce the number of candidates. Twenty-eight lncRNAs were excluded from further study, as their expression was below detection in more than half of the samples. After univariate Cox analysis, we adopted the least absolute shrinkage and selection operator (LASSO) method to construct a lncRNA signature for predicting NPC metastasis in the training cohort. We generated a risk score using a formula that included the nine selected lncRNAs weighted by their regression coefficients in the penalized Cox model as follows: Risk score = $(0.117 \times$ expression of lnc-TRAPPC6B-2) + $(0.042 \times$ expression of lnc-DRD5-10) + $(-0.054 \times$ expression of NR2F2-AS1) + $(0.077 \times$ expression of lnc-CETP-1) + $(0.121 \times$ expression of lnc-CDK1-1) + $(0.100 \times$ expression of LINC02065) + $(0.196 \times$ expression of lnc-POTEH-7) + $(-0.409 \times$ expression of lnc-STX6-2) + $(-0.109 \times$ expression of lnc-C11orf91-2). Then, an optimal cutoff value (1.007, AUC = 0.78) was selected via receiver operating characteristic (ROC) curve analysis.

**Performance of the nine-lncRNA signature for NPC metastasis**. With the optimal cutoff value, 51 patients in the training cohort were classified into the high-risk group, and the remaining 126 patients were classified into the low-risk group. Patients in the high-risk group had poorer distant metastasis-free survival (DMFS; HR 6.04, 95% CI 2.82–12.94, $P = 1.4e-07$), disease-free survival (DFS; HR 2.34, 95% CI 1.33–4.10, $P = 2.2e-03$) and overall survival (OS; HR 3.79, 95% CI 1.95–7.35, $P = 2.3e-05$) than those in the low-risk group (Fig. 2a–c). The number of patients who had an event for each risk group is listed in Supplementary Table 3.

Univariate analysis showed that the lncRNA signature was significantly associated with DMFS (Fig. 3). Other prognostic factors of DMFS included N stage and EBV DNA (N stage: HR 3.75, 95% CI 1.66–8.46, $P = 0.002$; EBV DNA: HR 3.61, 95% CI 1.48–8.84, $P = 0.005$; Fig. 3). Multivariate analysis revealed that

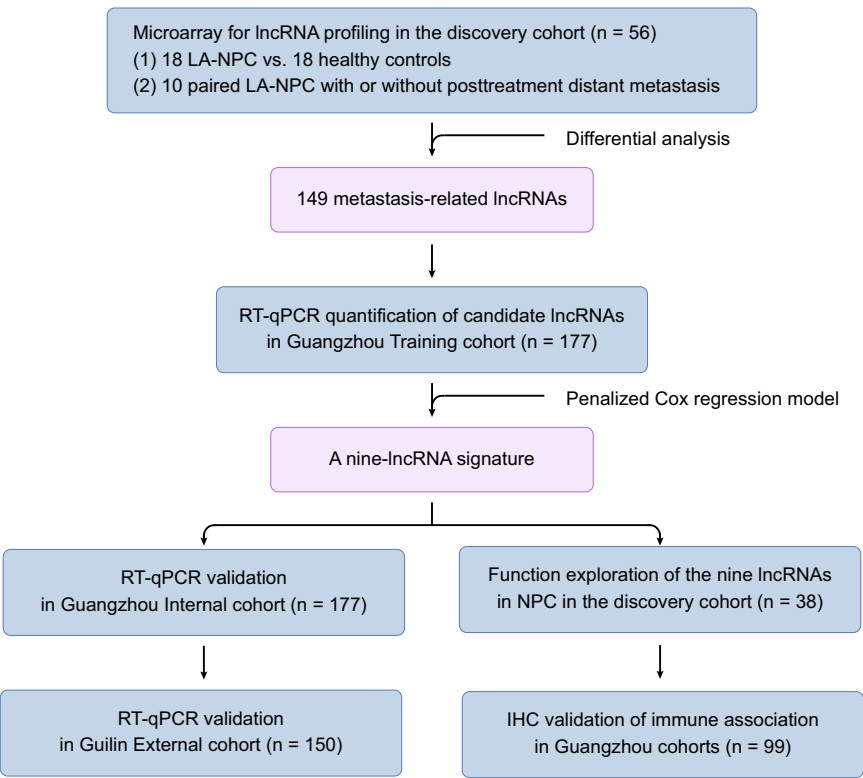

**Fig. 1 Study design.** This figure shows the workflow of the study. LA-NPC locoregionally advanced nasopharyngeal carcinoma, RT–qPCR quantitative reverse transcription PCR.

the nine-lncRNA signature, N stage and EBV DNA remained independent prognostic factors for DMFS (the nine-lncRNA signature: HR 5.10, 95% CI 2.36–11.01, $P = 3.5\text{e-}05$; N stage: HR 2.41, 95% CI 1.04–5.56, $P = 0.039$; EBV DNA: HR 2.86, 95% CI 1.15–7.08, $P = 0.023$; Supplementary Table 4).

**Validation of the nine-lncRNA signature in two independent cohorts.** To further confirm the value of the nine-lncRNA signature in identifying metastasis, we validated our findings in the other two independent cohorts, using the same formula and cutoff developed in the training cohort. We categorized 137 of 177 patients from the Guangzhou internal validation cohort into the low-risk group and 40 patients into the high-risk group, and the two groups were found to have significant differences in terms of DMFS, DFS, and OS (DMFS: HR 6.63, 95% CI 3.09–14.20, $P = 2.1\text{e-}08$; DFS: HR 4.05, 95% CI 2.23–7.34, $P = 6.1\text{e-}07$; OS: HR 5.39, 95% CI 2.71–10.73, $P = 7.8\text{e-}08$; Fig. 2d–f). Then, 32 out of 150 patients from the Guilin external validation cohort were classified into the high-risk group, which were found to have shorter DMFS, DFS, and OS (DMFS: HR 5.97, 95% CI 2.90–12.28, $P = 3.6\text{e-}08$; DFS: HR 2.74, 95% CI 1.58–4.76, $P = 1.9\text{e-}04$; OS: HR 3.87, 95% CI 1.90–7.87, $P = 5.6\text{e-}05$; Fig. 2g–i). The nine-lncRNA signature was subsequently confirmed as a prognostic factor for DMFS in the internal and external validation cohorts (Fig. 3). After multivariable adjustment, the nine-lncRNA signature remained a powerful independent indicator of DMFS (Guangzhou internal cohort: HR 6.71, 95% CI 3.12–14.43, $P = 1.1\text{e-}06$; Guilin external cohort: HR 6.38, 95% CI 3.09–13.18, $P = 5.3\text{e-}07$; Supplementary Tables 5, 6).

**The nine-lncRNA signature outperforms traditional clinical factors.** We then compared the performance of the nine-lncRNA signature with that of N stage and EBV DNA for the prediction of metastasis, as these factors were significantly associated with DMFS in univariate Cox analysis. The results of the ROC analysis showed that the nine-lncRNA signature demonstrated superior efficacy than the N stage or EBV DNA alone in the training and two independent validation cohorts (Fig. 4a–c).

In addition, the nine-lncRNA signature and N stage were independent predictors of NPC metastasis in multivariate analyses after adjustment by other clinical variables (Supplementary Tables 4–6). To develop a more accurate prognostic tool, we built a model by combining the nine-lncRNA signature and N stage in the training cohort (Supplementary Table 7). When compared with the N stage alone, the combined model significantly improved the efficiency in predicting metastasis in the training cohort ($\text{AUC}_{\text{N stage}} = 0.65$; $\text{AUC}_{\text{combined}} = 0.77$; $P = 0.005$, Fig. 4d), which was corroborated in two validation cohorts (Guangzhou internal validation cohort: $\text{AUC}_{\text{N stage}} = 0.63$; $\text{AUC}_{\text{combined}} = 0.80$; $P = 0.003$; Guilin external validation cohort: $\text{AUC}_{\text{N stage}} = 0.62$; $\text{AUC}_{\text{combined}} = 0.76$; $P = 0.007$; Fig. 4e, f).

**Biological characteristics related to the nine-lncRNA signature.** To explore the function of the nine lncRNAs in the signature, we performed functional enrichment analysis with the microarray data of LA-NPC samples from the discovery cohort. In addition to the pathways highly correlated with tumorigenesis and metastasis, such as Myc targets and epithelial-mesenchymal transition (EMT), many of the nine lncRNAs were also associated with immune-related pathways (Fig. 5a). To investigate this finding more deeply, we focused on 17 immunologically relevant gene sets derived from the ImmPort website[27]. The results further demonstrated that the nine lncRNAs were involved in immune pathways, especially those related to inflammatory cytokines, T cell receptor (TCR) signaling, and B cell receptor (BCR) signaling (Fig. 5b).

Next, we explored whether the nine-lncRNA signature correlated with immune function, similar to the nine individual

**Table 1 Patient characteristics in the training, internal and external validation cohorts stratified according to the immune-related lncRNA signature.**

| | Training cohort (n = 177) | | | | Guangzhou internal validation cohort (n = 177) | | | | Guilin external validation cohort (n = 150) | | | |
|---|---|---|---|---|---|---|---|---|---|---|---|---|
| | No. of patients | Low risk no. (%) | High risk no. (%) | P | No. of patients | Low risk no. (%) | High risk no. (%) | P | No. of patients | Low risk no. (%) | High risk no. (%) | P |
| Age | | | | 0.332 | | | | 0.323 | | | | 0.999 |
| <45 years | 102 | 76 (60.3) | 26 (51.0) | | 94 | 76 (55.5) | 18 (45.0) | | 50 | 39 (33.1) | 11 (34.4) | |
| ≥45 years | 75 | 50 (39.7) | 25 (49.0) | | 83 | 61 (44.5) | 22 (55.0) | | 100 | 79 (66.9) | 21 (65.6) | |
| Sex | | | | 0.948 | | | | 0.684 | | | | 0.590 |
| Male | 140 | 99 (78.6) | 41 (80.4) | | 126 | 96 (70.1) | 30 (75.0) | | 102 | 82 (69.5) | 20 (62.5) | |
| Female | 37 | 27 (21.4) | 10 (19.6) | | 51 | 41 (29.9) | 10 (25.0) | | 48 | 36 (30.5) | 12 (37.5) | |
| T stage | | | | 0.732 | | | | 0.806 | | | | 0.197 |
| T1 | 5 | 4 (3.2) | 1 (2.0) | | 8 | 7 (5.1) | 1 (2.5) | | 3 | 3 (2.5) | 0 (0) | |
| T2 | 16 | 10 (7.9) | 6 (11.8) | | 10 | 7 (5.1) | 3 (7.5) | | 25 | 21 (17.8) | 4 (12.5) | |
| T3 | 116 | 85 (67.5) | 31 (60.8) | | 110 | 84 (61.3) | 26 (65.0) | | 74 | 61 (51.7) | 13 (40.6) | |
| T4 | 40 | 27 (21.4) | 13 (25.5) | | 49 | 39 (28.5) | 10 (25.0) | | 48 | 33 (28.0) | 15 (46.9) | |
| N stage | | | | 0.086 | | | | 0.986 | | | | 0.514 |
| N0 | 16 | 13 (10.3) | 3 (5.9) | | 11 | 9 (6.6) | 2 (5.0) | | 10 | 9 (7.6) | 1 (3.1) | |
| N1 | 76 | 60 (47.6) | 16 (31.4) | | 78 | 60 (43.8) | 18 (45.0) | | 54 | 41 (34.7) | 13 (40.6) | |
| N2 | 51 | 33 (26.2) | 18 (35.3) | | 58 | 45 (32.8) | 13 (32.5) | | 66 | 54 (45.8) | 12 (37.5) | |
| N3 | 34 | 20 (15.9) | 14 (27.5) | | 30 | 23 (16.8) | 7 (17.5) | | 20 | 14 (11.9) | 6 (18.8) | |
| EBV DNA (copies/mL) | | | | 0.450 | | | | 0.973 | | | | |
| <2000 | 79 | 59 (46.8) | 20 (39.2) | | 69 | 54 (39.4) | 15 (37.5) | | / | / | / | |
| ≥2000 | 98 | 67 (53.2) | 31 (60.8) | | 108 | 83 (60.6) | 25 (62.5) | | / | / | / | |
| Distant metastasis | | | | 1.6e-06 | | | | 5.5e-07 | | | | 5.8e-06 |
| Yes | 30 | 10 (7.9) | 20 (39.2) | | 28 | 11 (8.0) | 17 (42.5) | | 30 | 14 (11.9) | 16 (50.0) | |
| No | 147 | 116 (92.1) | 31 (60.8) | | 149 | 126 (92.0) | 23 (57.5) | | 120 | 104 (88.1) | 16 (50.0) | |
| Disease progression | | | | 8.9e-03 | | | | 1.1e-05 | | | | 9.2e-03 |
| Yes | 50 | 28 (22.2) | 22 (43.1) | | 44 | 23 (16.8) | 21 (52.5) | | 57 | 38 (32.2) | 19 (59.4) | |
| No | 127 | 98 (77.8) | 29 (56.9) | | 133 | 114 (83.2) | 19 (47.5) | | 93 | 80 (67.8) | 13 (40.6) | |
| Death | | | | 1.7e-04 | | | | 3.6e-06 | | | | 7.0e-04 |
| Yes | 36 | 16 (12.7) | 20 (39.2) | | 33 | 15 (10.9) | 18 (45.0) | | 31 | 17 (14.4) | 14 (43.8) | |
| No | 141 | 110 (87.3) | 31 (60.8) | | 144 | 122 (89.1) | 22 (55.0) | | 119 | 101 (85.6) | 18 (56.2) | |

The P values were determined using the two-tailed χ² test. Source data are provided as a Source Data file.
TNM tumor-node-metastasis, EBV Epstein-Barr virus.

lncRNAs. By comparing gene expression in the high-risk and low-risk groups divided based on the lncRNA signature, we discovered that while genes upregulated in the high-risk group were primarily enriched in malignant property-related and metabolism-related pathways, genes upregulated in the low-risk group were mainly enriched in immune-related pathways, suggesting that the lncRNA signature may distinguish tumors with distinct immune heterogeneity (Fig. 5c). As expected, immune-related genes showed clear differences between the patients in different groups. We found that the low-risk group had significantly higher expression of immune-related genes, including those representing cytotoxic functions, chemokines and cytokines, antigen presentation, and immune cell infiltration (Fig. 5d).

**Association of the nine-lncRNA signature and lymphocyte infiltration.** As the genes representing different immune cell types were differentially expressed in the high- and low-risk group, we investigated whether the immune infiltration pattern also differed between the two groups. Immune infiltration was estimated by the MCP-counter algorithm with microarray data. By performing RT–qPCR assay, we calculated the risk score for LA-NPC patients in the discovery cohort to distinguish between high- and low-risk patients. Patients in the low-risk group had a greater level of B cell and CD8+ T cell infiltration (Fig. 6a). To validate the result of the bioinformatic analysis, we performed

hematoxylin and eosin (H&E) staining and immunohistochemistry (IHC) analyses. To prevent subjective interpretation of the pathologic results, we adopted a digital pathology method (Fig. 6b). The results revealed different immune infiltration in NPC tissues (Fig. 6c) and confirmed that B cells and intratumor CD8+ T cells were significantly more abundant in low-risk patients (Intratumoral CD20+ B cells, P = 0.011; stromal CD20+ B cells, P = 0.031; intratumoral CD8+ T cells, P = 0.045; Fig. 6d). Stromal CD8+ T cells showed a similar tendency but with marginal significance (stromal CD8+ T cells, P = 0.058; Fig. 6d).

**Discussion**

In this multicenter, retrospective cohort study, we identified an immune-associated signature in LA-NPC patients based on nine lncRNAs. The lncRNA signature improved the prognostic stratification of patients with LA-NPC by effectively distinguishing between patients with high- and low-risk of distant metastasis. Furthermore, the lncRNA signature outperformed N stage and EBV DNA and provided additional prognostic value to these existing predictors. Our findings demonstrate the utility of lncRNAs as a metastasis prediction tool in LA-NPC patients.

Distant metastasis is the primary reason for treatment failure in patients with NPC[4], and clinicopathologic factors show relatively poor performance in distinguishing patients at high risk[6]. Luckily, recent studies have revealed the diverse functions of lncRNAs in regulating cancer metastasis, suggesting that lncRNAs could serve

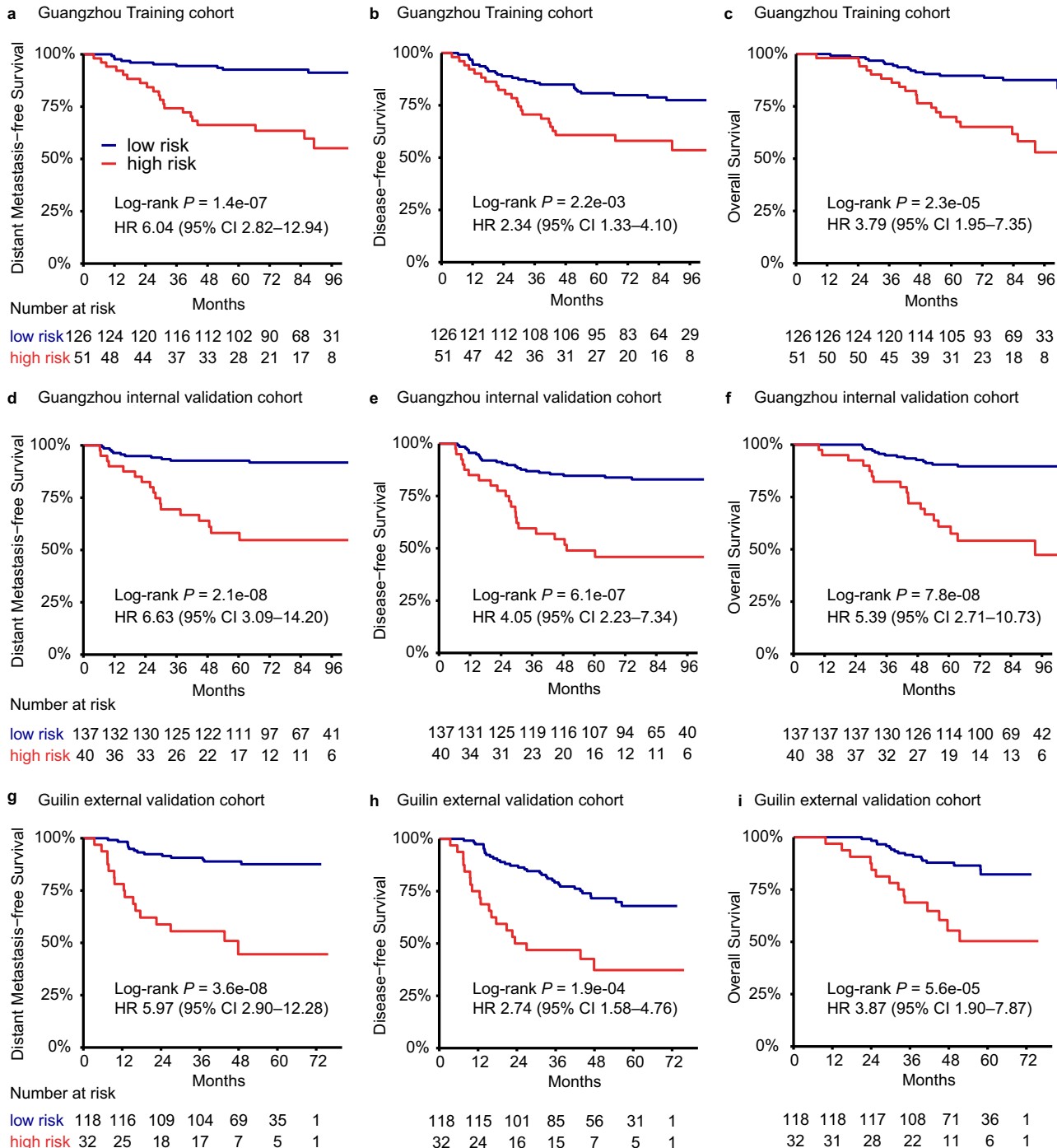

**Fig. 2 Kaplan–Meier estimates survival curves for the high-risk and low-risk groups according to the nine-lncRNA signature. a–c** Distant metastasis-free survival (**a**), disease-free survival (**b**), and overall survival (**c**) in the training cohort (n = 177). **d–f** Distant metastasis-free survival (**d**), disease-free survival (**e**), and overall survival (**f**) in the Guangzhou internal validation cohort (n = 177). **g–i** Distant metastasis-free survival (**g**), disease-free survival (**h**), and overall survival (**i**) in the Guilin external validation cohort (n = 150). The log-rank test was used to calculate P values (two-sided), and univariate Cox regression analyses were used to estimate the hazard ratios. HR hazard ratio, CI confidence interval. Source data are provided as a Source Data file.

as potential prognostic biomarkers for tumor patients[22]. However, many lncRNA biomarker studies have been conducted based on data from the public databases, and have not been subjected to large-scale, independent validation[25,26]. In the current study, we adopted a three-step strategy to develop and validate a lncRNA signature. Microarrays were used to detect global lncRNA expression in LA-NPC patients and metastasis-related lncRNAs were discovered in the discovery stage. In the training stage, we used RT–qPCR to quantify the selected

lncRNAs in a larger cohort, and developed a nine-lncRNA signature. Finally, we confirmed the performance of the nine-lncRNA signature in two independent validation cohorts. We believe that the proposed lncRNA signature could be a promising predictive tool for metastasis of NPC.

LncRNAs play significant roles in gene regulation and have emerged as critical regulators of cancer metastasis[17,18,28]. The functions of the nine lncRNAs in our signature are mostly unknown, except for that of NR2F2-AS1 and lnc-POTEH-7,

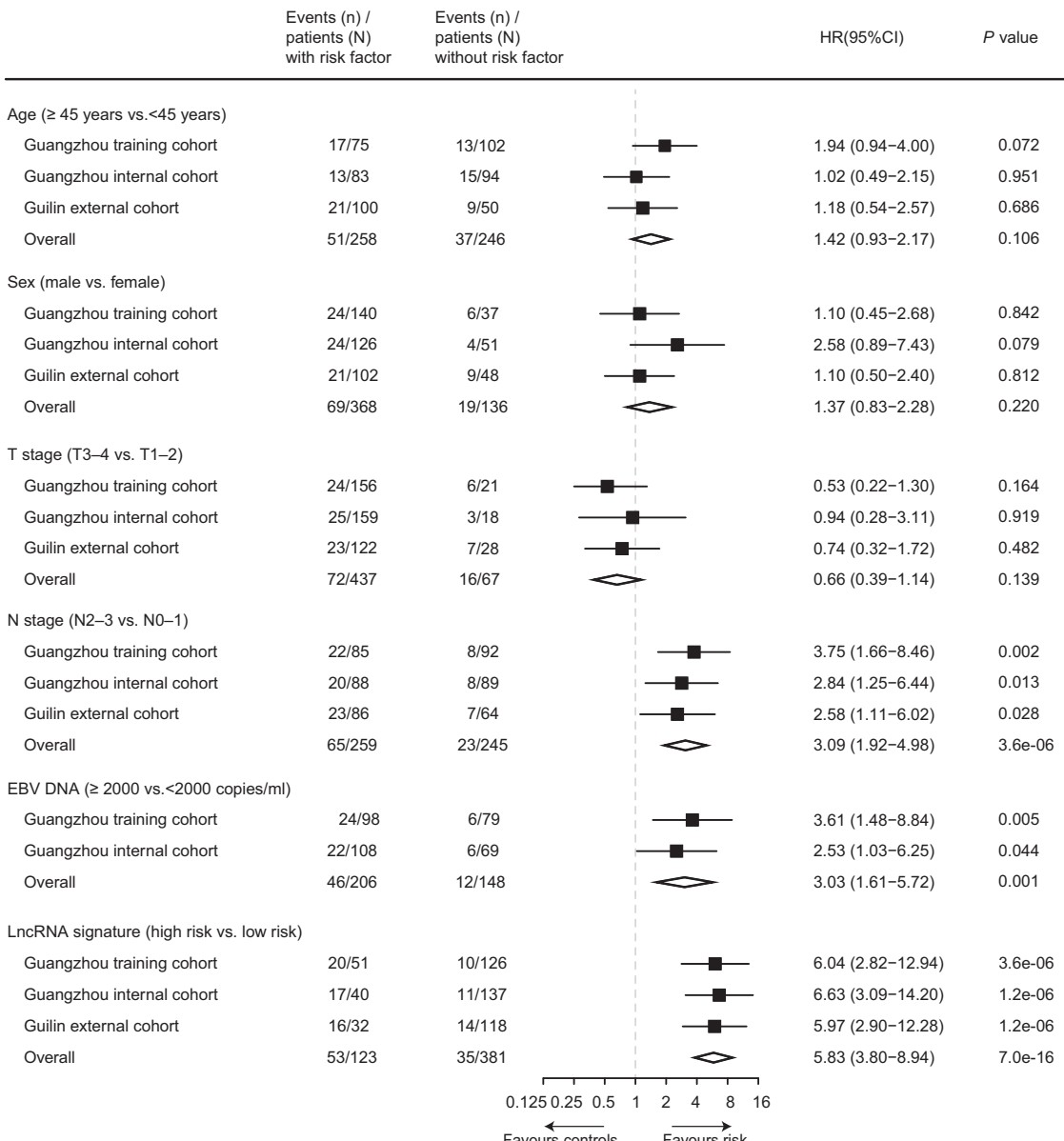

**Fig. 3 Forest plots of the nine-lncRNA signature and clinicopathological characteristics on distant metastasis-free survival.** Hazard ratios (HR), 95% confidence interval (CI), and *P* values were calculated by an unadjusted Cox proportional-hazards model with a two-sided Wald test. Squares represent the hazard ratios with error bars corresponding to 95% CI bounds. EBV Epstein-Barr virus, HR hazard ratio, CI confidence interval. Source data are provided as a Source Data file.

which have been reported to promote tumor invasion in multiple cancers[29–32]. Therefore, we explored the function of these lncRNAs with functional enrichment analysis. Interestingly, the results showed that these lncRNAs are probably related to immune function. Recent studies have discovered that lncRNAs are crucial regulators of the differentiation and function of immune cells that significantly affect the TME and ultimately contribute to tumor metastasis[33,34]. In light of this finding, we further analyzed differences in the level of immune infiltration in the high- and low-risk groups as defined by the lncRNA signature, and the results suggested that high-risk patients had deficient CD8$^+$ T cell and B cell infiltration. In other words, this lncRNA signature is likely indicative of immune infiltration. Inspired by this result, we further collected available paraffin-embedded samples in Guangzhou cohorts and detected CD8$^+$ T cell and B cell infiltration, and the analysis yielded similar results. These findings indicated that our lncRNA signature could

effectively classify LA-NPC patients into high- and low-risk groups, perhaps based on the immune association of these lncRNAs.

This immune-associated lncRNA signature displayed better performance than the N stage with regard to predicting distant metastasis in LA-NPC. The reason might be that while the N stage reflects anatomical differences, the lncRNAs reflect the biological heterogeneity and immune diversity of tumors, providing deeper insights into a patient's systemic status. Next, we proposed that the addition of the immune-associated lncRNA signature to the N stage may improve the prediction accuracy. Indeed, the combination of N stage and the immune-associated lncRNA signature further boosted the predictive accuracy. Thus, our signature provides a simple and effective method for improved patient stratification.

Our present study has several strengths. First, we developed an immune-associated lncRNA signature based on RT–qPCR data

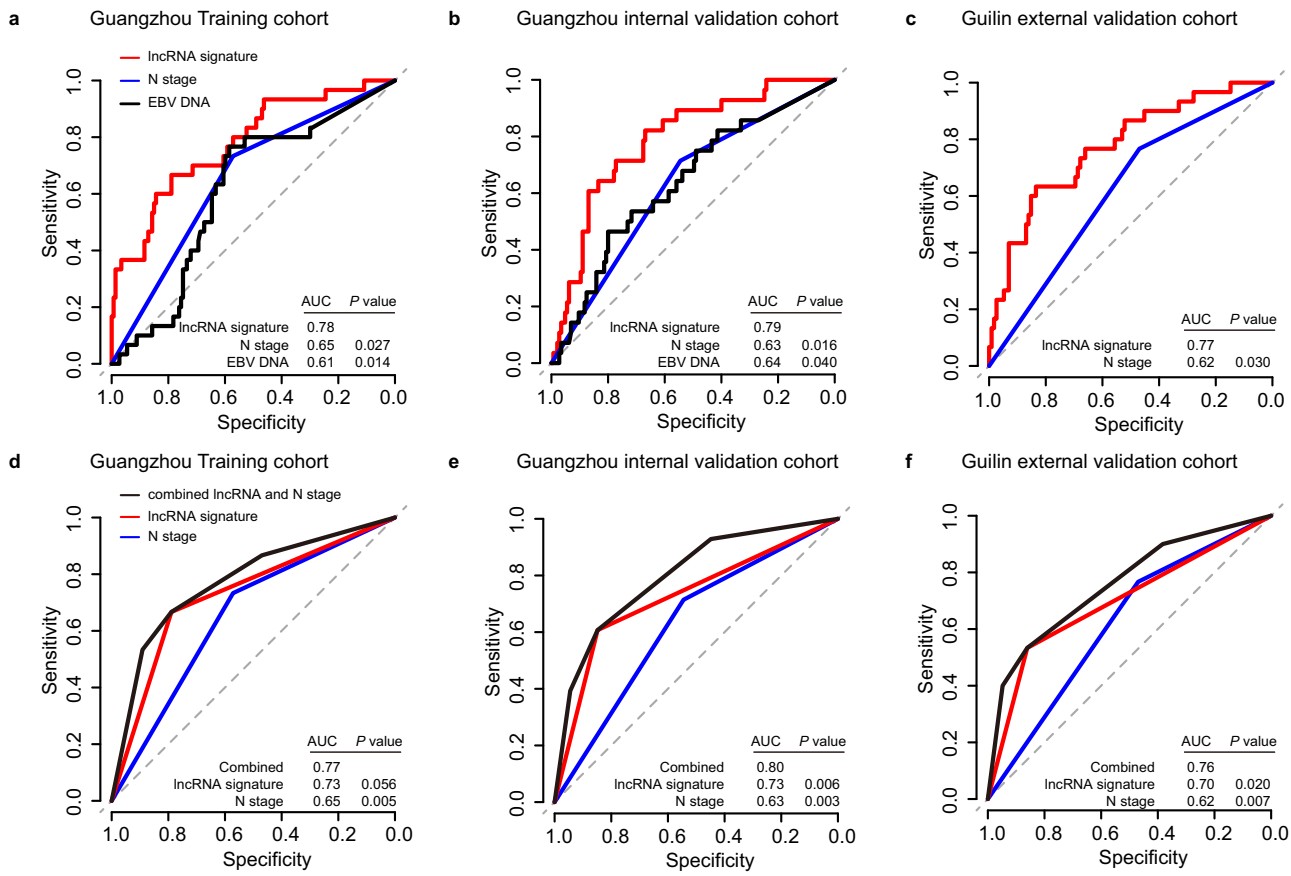

**Fig. 4 Performance of the nine-lncRNA signature and clinical indicators for metastasis prediction. a–c** Receiver operating characteristic (ROC) curve analysis evaluating the performance of the lncRNA signature, N stage, and EBV DNA for the prediction of distant metastasis in patients in the training (**a**, $n = 177$), Guangzhou internal validation (**b**, $n = 177$), and Guilin external validation (**c**, $n = 150$) cohorts. **d–f** ROC curve analysis evaluating the performance of the combined model (the lncRNA signature and N stage), the lncRNA signature alone and N stage alone for the prediction of distant metastasis in patients in the training (**d**, $n = 177$), Guangzhou internal validation (**e**, $n = 177$), and Guilin external validation (**f**, $n = 150$) cohorts. Two-sided DeLong's test was used to estimate the $P$ values. EBV Epstein-Barr virus, AUC area under curve. Source data are provided as a Source Data file.

that is feasible to use in most hospitals and thus has great practical value in clinical settings. In addition, we subjected our signature to multicenter validation, proving the effectiveness of this promising tool. Moreover, the potential immune association of these lncRNAs was suggested by bioinformatic analyses and validated by digital pathology, although the sample size was limited. Finally, we acknowledge that this study was limited by its retrospective nature. These findings require confirmation in large-scale prospective studies.

In conclusion, we developed an immune-associated lncRNA signature consisting of nine lncRNAs that performed much better than clinical indicators in predicting the distant metastasis of LA-NPC. The signature is easily formulated using expression data quantified by RT–qPCR assay and may serve as a valuable tool for classifying patients with different levels of metastatic risk and could aid in providing more individualized treatment to LA-NPC patients.

## Methods

**Patient population and study design**. This study was approved by the Institutional Ethical Review Boards of SYSUCC and the Affiliated Hospital of Guilin Medical College. The informed consent was obtained from all patients. We retrospectively collected fresh frozen tissue samples from 542 patients with stage III–IVa LA-NPC and 18 healthy controls. Samples were obtained from 392 LA-NPC patients and 18 healthy controls at Sun Yat-sen University Cancer Center (SYSUCC; Guangzhou, China) between July 2010 and December 2016. Among them, 38 samples were obtained from LA-NPC patients and 18 samples were obtained from healthy controls and were assigned to the discovery cohort;

354 samples obtained from LA-NPC patients were randomly classified into the training cohort ($n = 177$), and the Guangzhou internal validation cohort ($n = 177$). An additional 150 samples obtained at the Affiliated Hospital of Guilin Medical College (Guilin, China) between September 2014 and October 2017 were used as an external validation cohort (Fig. 1). Moreover, paraffin-embedded tumor samples were available from 99 LA-NPC patients in the Guangzhou training and validation cohorts.

All patients were pathologically diagnosed with NPC and restaged according to the 8th edition of the American Joint Committee on Cancer (AJCC) staging system[35]. All patients received platinum-based chemotherapy and IMRT[36]. No patients had received any antitumor therapy before sampling. All plasma EBV DNA levels in the Guangzhou cohorts were measured in the laboratory of the Department of Molecular Diagnosis at SYSUCC.

In this study, we first profiled lncRNA expression using high-throughput microarrays in the discovery cohort to screen candidate lncRNAs related to NPC distant metastasis. We then detected metastasis-related lncRNAs with RT–qPCR assays in the training cohort ($n = 177$) and developed a lncRNA signature for metastasis prediction in LA-NPC, which was validated in the Guangzhou internal ($n = 177$) and Guilin external ($n = 150$) validation cohorts. To further explore the potential function of the lncRNA signature, we employed functional enrichment analysis based on the guilt-by-association principle[37] and performed microenvironment cell populations (MCP)-counter immune estimation[38] with microarray data of the discovery cohort. The results were further confirmed with a digital pathology method utilizing samples from the Guangzhou cohorts ($n = 99$) (Fig. 1).

**Microarray analysis**. RNA was extracted using the AllPrep DNA/RNA Mini Kit (QIAGEN GmbH, Hilden, Germany). RNA concentration was measured by NanoDrop ND-1000 and RNA integrity was assessed by standard denaturing agarose gel electrophoresis. LncRNA profiles were detected using the Arraystar Human LncRNA Microarray[39] (V3.0, Agilent Technologies, Santa Clara, CA, USA), which was designed for the global profiling of 30,586 human lncRNAs and

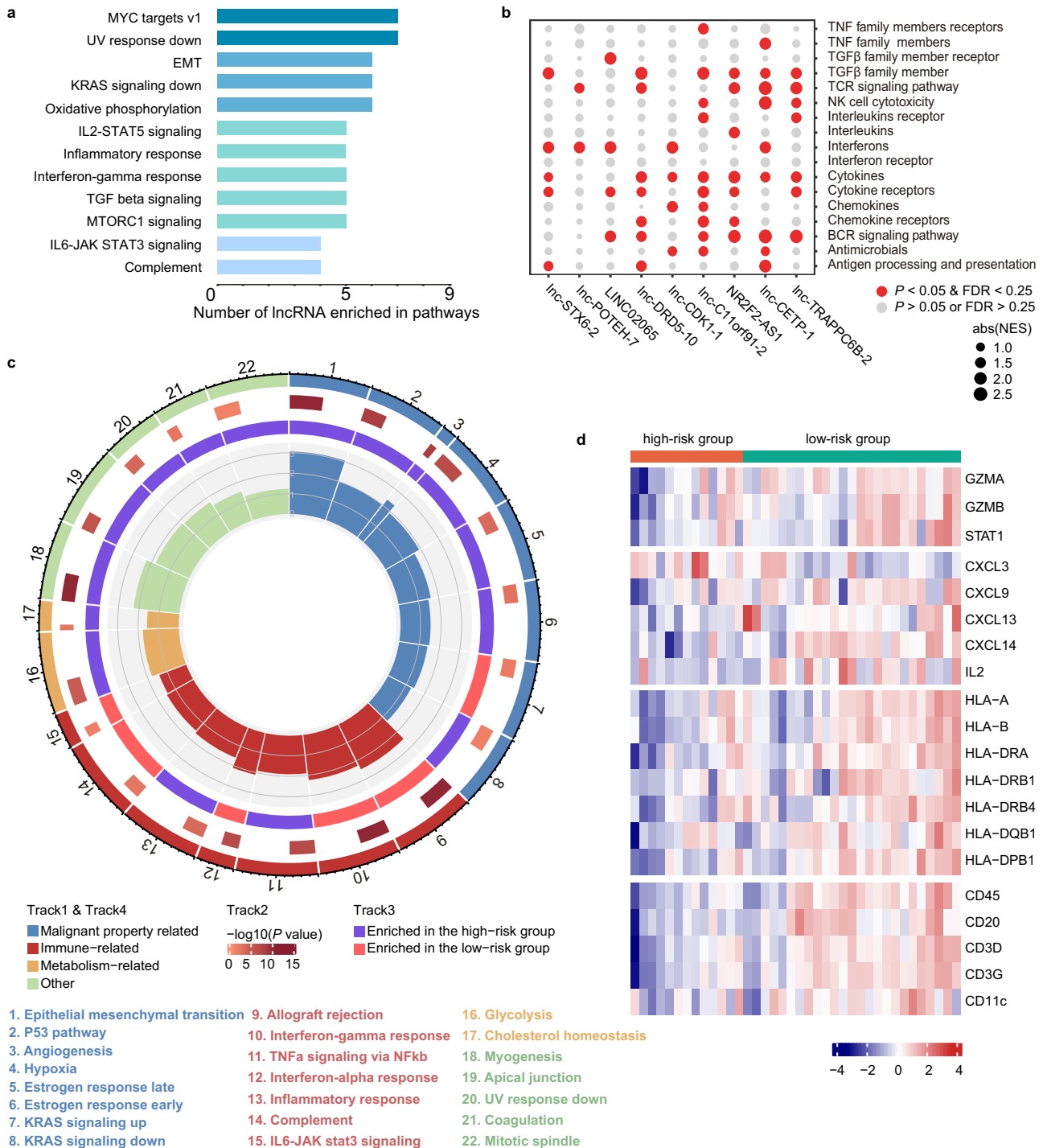

**Fig. 5 Molecular characteristics of high-risk and low-risk groups. a** Bar plot showing the number of lncRNAs out of the nine lncRNAs in the signature that are enriched in that corresponding pathway by functional enrichment analyses based on GSEA. Permutation-based *P* value shows the statistical significance of the normalized enrichment score (NES). Adjustments for multiple comparisons were presented by false discovery rate (FDR). Significantly enriched pathways are defined as *P* < 0.05, FDR < 0.25, and absolute NES > 1. **b** Bubble plot of GSEA results of the nine lncRNAs with immunologically relevant pathways in LA-NPC patients of the discovery cohort (*n* = 38). The red color of the dots represents enrichment (*P* < 0.05, FDR < 0.25, absolute NES > 1), and the size of the dots represents absolute NES. Permutation-based *P* value shows the statistical significance of NES and adjustments for multiple comparisons were presented by FDR. **c** Circos plot showing the significantly enriched pathways between the high- and low-risk groups based on GSEA. The size of each sector represents the number of genes in the labeled pathway. The color of the outmost circle and inner bar plots represent the category of pathways. The size of the second outer circle represents the percentage of genes contributing to the enrichment score, and its color represents the magnitude of the statistical significance (shown as −log10 (*P* value)). The color of the third circle indicates that the labeled pathway is upregulated in the high- or low-risk group. The size of the inner bar plot shows the absolute NES. Permutation-based *P* value shows the statistical significance of NES and adjustments for multiple comparisons were presented by FDR. **d** Heatmap of differentially expressed immune-related genes between the high- and low-risk groups (Student's two-sided *t*-test, *P* value < 0.05). FDR false discovery rate, NES normalized enrichment score. Source data are provided as a Source Data file.

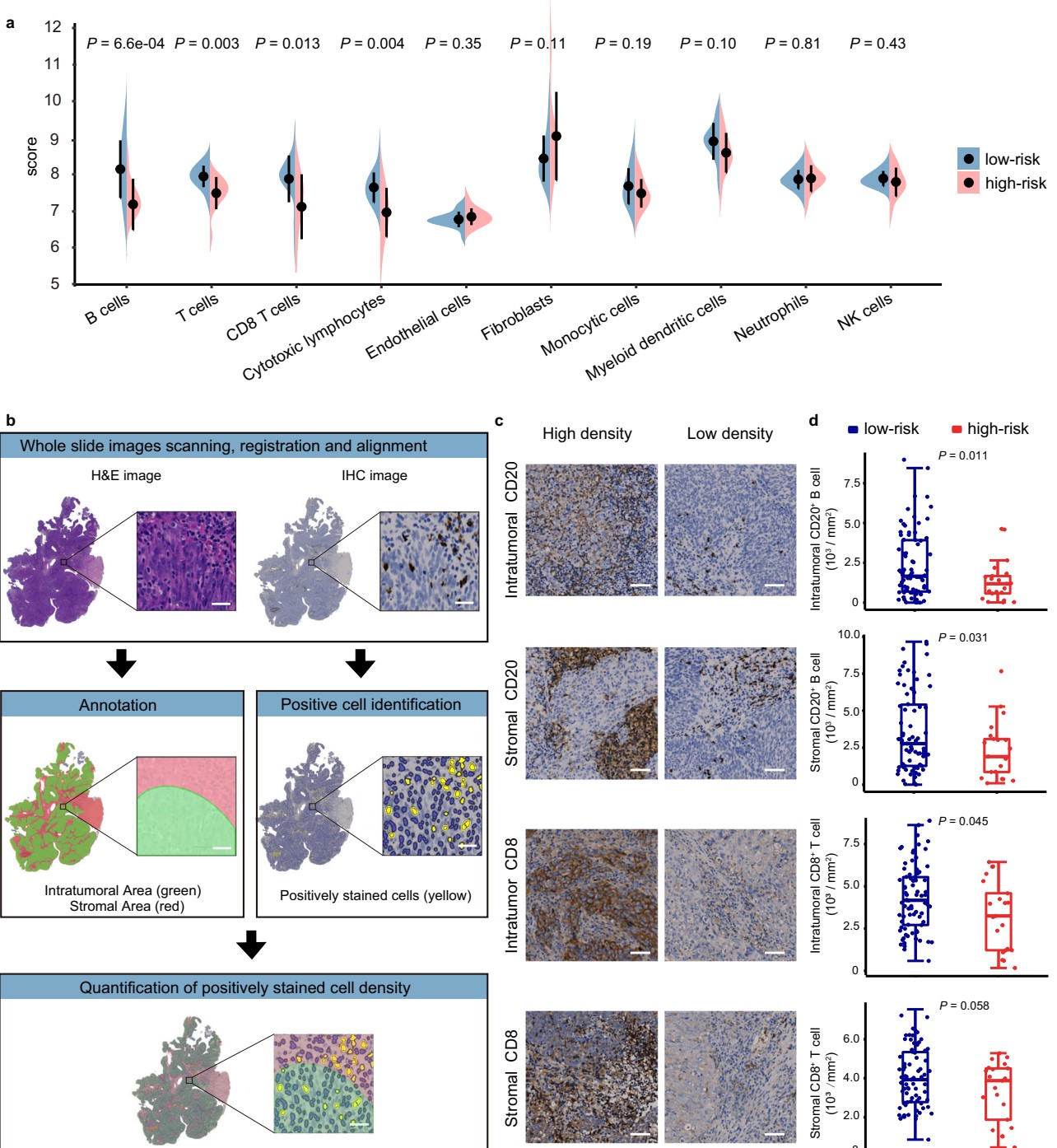

**Fig. 6 Immune infiltration of high-risk and low-risk groups. a** Immune infiltration of patients (high-risk, $n = 13$; low-risk, $n = 25$) estimated by the MCP-counter algorithm. The dots represent the mean scores, and the error bars represent the standard deviation. The comparison was based on Student's two-sided $t$-test. **b** Digital pathology analysis pipeline. For each sample, hematoxylin and eosin (H&E) and immunohistochemistry (IHC) slides were scanned, registered, and aligned in HALO software. Manual annotation of tumor nests and stroma areas on the H&E image were automatically synchronized to the IHC image. Positively stained cells in the IHC images were identified by the Multiplex IHC algorithm. Quantification of positively stained cells and the areas of tumor nests and stroma were automatically generated by HALO software. The scale bar represents 25 μm. **c** Representative images of CD20 and CD8 expression in intratumoral and stromal areas of tumor sections detected by IHC analysis in the high-risk ($n = 21$) and low-risk groups ($n = 78$). The images show high (left panel) or low (right panel) CD20+ B and CD8+ T cell densities in intratumoral and stromal areas, respectively. The scale bar represents 50 μm. **d** Comparison of CD20- and CD8-positive cells in intratumoral and stromal areas in patients in the high-risk ($n = 21$) and low-risk groups ($n = 78$) stratified by the nine-lncRNA signature. In each box plot, the centerline represents the median, the bounds represent the first and third quartiles and whiskers extend from the hinge to the largest value no further than 1.5 * interquartile range (IQR) from the hinge. Each dot represents a data point for individual patients. The comparison was based on Student's two-sided $t$-test. Source data are provided as a Source Data file.

26,109 protein-coding transcripts. After sample labeling and array hybridization, raw data were extracted using the Agilent Feature Extraction software (version 11.0.1.1). Quantile normalization, quality control, and differential expression analyses were performed with GeneSpring GX v12.1 software (Agilent Technologies). After normalization, transcript-specific probes for lncRNA labeled as "present" in more than or equal to half of the samples were considered high-qualified for the biomarker selection. Differential expression analyses were performed based on t-test between samples from 18 LA-NPC and 18 matched healthy controls, as well as between ten paired samples from LA-NPC developed with or without posttreatment distant metastasis. The lncRNAs showing significant differential expression between the two groups were identified with cutoffs of $P < 0.05$ and fold change >1.5. For integrative lncRNA and mRNA analysis, we only included samples from LA-NPC and removed the batch effect using the combat algorithm in "sva" package[40] in R software. Protein-coding probes with flags "present" or "marginal" in more than or equal to ten samples were incorporated into the analysis.

**SmartChip qPCR analysis**. Total RNA was reverse-transcribed using the GoScript™ Reverse Transcription System (Promega Corporation, Madison, Wisconsin, USA). The complementary DNA was analyzed by qPCR using Invitrogen™ Platinum™ SYBR Green SuperMix-UDG reagents (Thermo Fisher) with the Wafergen SmartChip Real-time PCR system (Takara Bio USA, Inc., San Jose, CA). This system is suitable for large-scale gene expression studies, which can process 5184 nanowell reactions per run. After the initial denaturation at 95 °C for 5 min, 45 cycles of the following program were used for amplification: 95 °C for 15 s, 60 °C for 15 s, and 72 °C for 15 s. The melting curve analysis was automatically generated by Wafergen SmartChip qPCR analysis software (version 2.8.6.1). After that, raw data of qPCR were also obtained with Wafergen SmartChip qPCR analysis software, excluding several nanowells with multiple melting peaks or amplification efficiency less than 1.50. LncRNA expressions were measured using the ΔCt method with *ACTB* as the internal control gene (lncRNA Cq value subtracts *ACTB* Cq value within the same sample). The PCR primers used in this study were listed in Supplementary Table 8.

**Construction of the nine-lncRNA signature**. To construct a lncRNA signature, we first used univariate Cox analysis to select lncRNAs related to DMFS in the training cohort. LASSO[41] was adopted to further reduce the number of candidates by the "glmnet" package[42] in R software. In short, a λ value was chosen via min (minimum error) criteria under ten-fold cross-validation. Based on the λ value, we could select the lncRNAs whose beta coefficients were not zero to calculate a risk score for each patient. The risk score was generated using a formula derived from the expression levels of these lncRNAs weighted by their coefficients and then a ROC curve[43] was used to determine the optimal cutoff values with the "pROC" package[44] in R software. Patients were divided into low- and high-risk groups with the threshold that produced the maximum sum of sensitivity and specificity in the ROC curve.

**Functional enrichment analysis**. We investigated the functional contexts of the nine lncRNAs in our signature based on the guilt-by-association principle[37]. This approach is based on correlation analysis between matching lncRNA and mRNA expression in combination with enrichment strategies. It contains three steps: (i) for each lncRNA, an expression-correlation matrix (Spearman's rank) was constructed by integrating lncRNA and matching mRNA expression data; (ii) the expression-correlation matrix was ranked by correlation coefficient in descending order; and (iii) the sorted expression-correlation matrix was used as input for GSEA (hallmark gene set list obtained from MSigDb and immune-related pathways from the ImmPort website[27]) with "clusterProfiler" package[45] in R software. Pathways with $P < 0.05$, false discovery rate (q-value) <0.25 and absolute normalized enrichment score (|NES|) >1 were considered significant and were used to infer the functions of the corresponding lncRNAs.

**Digital pathology analysis**. Sequential tumor sections were used for H&E staining and IHC analyses. We chose anti-CD8 (ab4055, 1:800; Abcam) and anti-CD20 (HPA014341, 1:3000; Sigma Aldrich, Merck) antibodies as primary antibodies for IHC staining to detect CD8+ cytotoxic T cells and CD20+ B cells. Briefly, the tissues were deparaffinized and rehydrated, followed by blocking the endogenous peroxidase activity and citrate-mediated high-temperature antigen retrieval. After blocking the nonspecific binding, the samples were incubated with the primary antibodies at 4 °C overnight and labeled with HRP rabbit/mouse secondary antibodies (Dako REAL™ EnVision™), stained with diaminobenzidine (Sigma), and counterstained with hematoxylin[46,47]. All full view of each H&E and IHC sections were digitally scanned using a ZEISS Axio Scan.Z1 microscope at ×200 magnification and analyzed using Multiplex IHC algorithm[48,49] (version 3.1.4) implemented in HALO image software[49,50] (Indica Labs, USA). For each H&E slide, tumor and stromal areas in the whole tissue were manually annotated by two experienced pathologists according to criteria described by the International Immuno-Oncology Biomarker Working Group[51–53]. Registration and alignment of serial sections using the "Landmarks" function synchronized annotations on H&E sections to sequential IHC sections[54]. A training dataset for CD8- or CD20-stained images was established by randomly selecting 30 slides and performing nuclei segmentation and cell classification with the HALO multiplex IHC algorithm.

Automated classification of the positive staining cells was performed based on cytonuclear features such as stain intensity, size, and roundness. The trained algorithm was evaluated through visual inspection by pathologists and then applied to the rest of the slides with the same parameters. Finally, the algorithm automatically quantified the numbers of positive cells and the areas of tumor and stroma across the whole slide image, and the results were presented as cell density per mm².

**Statistical analysis**. The primary endpoint of our study was DMFS, and the secondary endpoints were DFS and OS. We defined DMFS as the period from the first date of treatment to the date of first distant relapse; DFS as the period from the first date of treatment to the date of first relapse at any site or death from any cause, whichever occurred first; and OS as the period from the first date of treatment to the date of death from any cause.

We applied univariate Cox regression analysis to distinguish clinical features associated with clinical prognosis, and multivariate Cox regression analysis was performed to select independent prognostic factors. The Kaplan–Meier method and the log-rank test were used to estimate the survival probability of patients among different groups, and Cox regression analyses were applied to calculate the hazard ratios (HRs). ROC curves were generated to test the efficiency of our model and other factors. The $\chi^2$ test or Fisher's exact test was used to compare categorical variables. All statistical tests were performed in R software (version 4.0.3) with two-tailed tests, and a $P$ value < 0.05 was considered significant.

**Reporting summary**. Further information on research design is available in the Nature Research Reporting Summary linked to this article.

## Data availability

The publicly available hallmarks gene lists used in this study are available in the Molecular Signatures Database (MSigDB) database (http://www.gsea-msigdb.org/gsea/index.jsp), and the immune-related gene lists are available on the ImmPort website (https://www.immport.org/home). The microarray data used in this study have been deposited at Gene Expression Omnibus under accession code GSE180272. The remaining data are available within the Article, Supplementary Information, or Source Data file. Source data are provided with this paper.

## Code availability

Essential scripts for model development and validation in multiple cohorts are available at the Github website (https://github.com/YL-L26/lncRNA_signature_for_NPC)[55].

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

## Acknowledgements

The authors thank Dr. Zhang Yu (from the Department of Pathology, Sun Yat-sen University Cancer Center) for her assistance during the digital pathology analysis. This work was supported by grants from the National Key R&D Program of China (2021YFA1302100 to N.L.), the National Natural Science Foundation of China (81922057 to N.L.; 81930072 to JM), the Key-Area Research and Development Program of Guangdong Province (2019B020230002 to J.M.), the Natural Science Foundation of Guangdong Province (2017A030312003 to J.M.), and the Overseas Expertise Introduction Project for Discipline Innovation (111 Project, B14035 to J.M.).

## Author contributions

Conception and design: N.L., Y.-Q.L., J.M., and W.J.; Development of methodology: Y.-L.L., Y.Z., X.-R.T., H.Q., and S.-R.L.; Acquisition of data: Y.-L.L., Y.Z., N.L., Y.-Q.L., W.J., L.-L.T., Y.-P.M., L.C., W.-F.L., G.-Q.Z., Y.Z., J.-Y.L., Q.L., S.-Y.H., S.G., Z.-Q.Z., Z.-X.L., and Y.S.; Analysis and interpretation of data: Y.-L.L., Y.Z., X.-R.T., H.Q., S.-R.L., N.L., Y.-Q.L., J.M., and W.J.; Writing the manuscript: Y.-L.L., X.-R.T., N.L., Y.-Q.L.; Revision of the manuscript: all authors; Administrative, technical, or material support: J.M. and N.L.; Study supervision: N.L., Y.-Q.L., J.M., and W.J. All authors reviewed and approved the final manuscript.

## Competing interests

The authors declare no competing interests.
