## [Peer Review File · Nature Communications]

A lncRNA signature associated with tumor immune heterogeneity predicts distant metastasis in locoregionally advanced nasopharyngeal carcinomaEditorial Note: Parts of this Peer Review File have been redacted as indicated to remove third-party material where no permission to publish could be obtained.

REVIEWER COMMENTS

Reviewer #1 (Remarks to the Author):

It is a multicentre, retrospective study to identify a nine-lncRNA signature for predicting metastasis in locoregionally advanced NPC. The candidate lncRNAs were identified by the differential expression analysis using microarray. The lncRNA signature was established in the training cohort by LASSO, and further validated in two independent NPC cohorts, indicating the robustness and soundness of the analysis methods. This signature appears to be associated with immune activity and tumour lymphocyte infiltration, suggesting the importance of tumor microenvironment for NPC metastasis. The study is novel and original with high translational values. The reviewer has a few major and minor comments and suggestions to further improve the quality of the study.

Major comments/questions

1. The patient characteristics are summarized in Table 1. But it is unclear whether all the patients included in the study are sporadic? Is there any record for family history for all the patients in this study? The Guilin external validation cohort has more cases >45y. What is the reason for the age discrepancy between the internal and external cohorts? The p value for evaluating the association of risk groups characterized by the lncRNA signature and the clinical parameters should be added in this table. Are these lncRNAs correlated with the clinical parameters at the individual level?
2. Has the selected differentially expressed (DE) lncRNAs been validated by qPCR in the discovery cohort? What is the analysis power to identify DE lncRNAs in the discovery cohort?
3. Are these nine lncRNAs expressed in the cancer cells? It is worthwhile to identify the source of lncRNA expression detected in this study.
4. In Figure 6, the difference between the high- and low-risk groups was observed in the B cells and CD8+ T cell infiltration. Can these two cell populations be the predictors for NPC metastasis as well?
5. It is unclear whether the prediction accuracy can be further boosted by combining three variables (lncRNA signature, pre-EBV and N stage) together in Guangzhou training and validation cohorts. Both cohorts have the pre-EBV data.
6. The Cox proportional hazards model was used in the analysis. How to evaluate the assumption of the proportional hazard for the Cox model is valid in the analysis?
7. What is the possible reason for upregulation/downregulation of these lncRNA expression in NPC tissues?
8. A digital pathology method was used in the IHC analyses in this study. The details about how to verify the results from the digital analysis by HALO software should be added in the methods.

Minor comments

1. The font size for the pathways in Figure 5C is too small.
2. The circo plot in Figure 5C is difficult to read. What does the size of pathways 1-22 mean in the plot? Why the size of the pathway 3 "Angiogenesis" is much smaller than the pathway 1 "EMT"? More details for the figure legend should be added.

Reviewer #2 (Remarks to the Author):

A lncRNA signature associated with tumour immune heterogeneity predicts distant metastasis in locoregionally advanced nasopharyngeal carcinoma

The authors have identified a set of 9 long-non-coding RNAs to be predictive of locoregionally advanced nasopharyngeal carcinoma. The authors derive a weighted formula based on the expression levels (by qRT-PCR) of these 9 lncRNAs and show that it is predictive of LA-NPC metastasis in two cohorts of patients. The authors also show that the 9 lncRNAs are independently prognostic relative to other known prognostic factors and correlate with the level of immune infiltration.

Comments:

Introduction: The language needs overall work (academic style writing).

Line 198-215: Does the derived risk score weights/formula sensitive to the method/kit used for qRT-PCR? If the authors propose this model to be used by other investigators, it is important to test its sensitivity outside the developed lab.

Line 261: Please add more details of this analysis in the methods section. It is not clear how the correlation was performed, cut-off used, and so on.

Line 292: Can the authors explain this analysis further in methods?

- Was that tissue considered as a whole or only tumor-specific regions were used in the analysis?
- Was the total size of the tissue taken into consideration and normalized prior to showing a significant difference in immune T/B cells between patients?

Minor:

Line 83: Please provide a reference.

Line 87: The previous lines (as written) do not "demonstrate" lncRNAs as promising biomarkers. Just rewording to be suggestive with improving the overall flow.

Minor Typos/misc

Line 82: extra space while citing 8

Line 84: extra space after citing 9-11

Line 87: extra space while citing 16

Line 91: Did the authors mean high throughput?

Line 94: Please check grammar

Line 97: unavailability?

Line 268: Please cite Immport

Reviewer #3 (Remarks to the Author):

In this work, the authors integrated multiple NPC data and identified nine lncRNA related to prognosis through a series of conventional bioinformatics methods. The authors developed a nine-lncRNA module which were crucial to the NPC patients' prognosis and metastasis and validated the module in other independent data to prove the accuracy. The work prioritized lncRNA biomarker in NPC and provides a novel prognosis module for NPC subtype. However, there are some issues need to be solved.

Major Point:

1. The description of differential expression analysis is indistinct. The author should add specific methods or the R package. Besides, the author can provide two Supplementary table containing the detail result, such as the p value, FDR value for each gene.
2. Although lncRNA is very important in the process of disease development, protein coding genes contain a wider range of genetic information. Why does the author only analyze the prognosis factor based on lncRNA expression profile, rather than integrate the two expression profiles?
3. In the process of model verification, the author uses the same threshold. This value is judged according to the ROC curve. Why not recalculate the AUC in the validation data and select the corresponding threshold?
4. In the result 'Biological characteristics related to the nine-lncRNA signature', the description of correlation analysis is relatively simple. Which method does the author use? This should be described.

Minor Point:

1. The AUC value related to the optimal cutoff value should be showed in the result.

Dear Editor and Reviewers,

Thank you very much for your important and supportive comments, questions and suggestions, which have greatly helped us to improve our study. Enclosed is the revised version of manuscript (Ref.: NCOMMS-21-29177) entitled, "A lncRNA signature associated with tumor immune heterogeneity predicts distant metastasis in locoregionally advanced nasopharyngeal carcinoma". We have revised and modified the manuscript in accordance with the reviewers' comments and re-submitted the revised manuscript. We all appreciate your support and efforts on our manuscript and look forward to your further decision, and we sincerely hope to have the opportunity to publish this paper in *Nature Communications*.

A copy of the manuscript indicating where revisions have been made is included in the resubmission. The revisions are indicated using the "Track Changes" function in Word. Please find below our responses to the revisions proposed by the reviewers.

Response to Reviewer #1:

It is a multicentre, retrospective study to identify a nine-lncRNA signature for predicting metastasis in locoregionally advanced NPC. The candidate lncRNAs were identified by the differential expression analysis using microarray. The lncRNA signature was established in the training cohort by LASSO, and further validated in two independent NPC cohorts, indicating the robustness and soundness of the analysis methods. This signature appears to be associated with immune activity and tumor lymphocyte infiltration, suggesting the importance of tumor microenvironment for NPC metastasis. The study is novel and original with high translational values. The reviewer has a few major and minor comments and suggestions to further improve the quality of the study.

Major comments/questions

1. The patient characteristics are summarized in Table 1. But it is unclear whether all the patients included in the study are sporadic? Is there any record for family history for all the patients in this study? The Guilin external validation cohort has more cases >45y. What is the reason for the age discrepancy between the internal and external cohorts? The p value for evaluating the association

of risk groups characterized by the lncRNA signature and the clinical parameters should be added in this table. Are these lncRNAs correlated with the clinical parameters at the individual level?

Answer:

Thank you very much for your valuable comments.

(1) In light of the reviewer's comments, we investigated the family history of all patients with nasopharyngeal carcinoma (NPC) in this study, and information was available for patients in the Guangzhou training and internal validation cohorts. In total, 38 of 354 (10.7%) patients had a family history of NPC (parents or siblings). There were no differences in family history of NPC between the low- and high-risk groups (χ^2 test, **Table 1**).

Table 1. Family history of patients in the training and internal validation cohorts stratified according to the immune-related lncRNA signature

	Training cohort (n=177)			Guangzhou Internal validation cohort (n=177)		
	Low risk (n)	High risk (n)	P	Low risk (n)	High risk (n)	P
			0.82			0.99
Familial NPC	14	7		13	4	
Sporadic NPC	112	44		124	36	

(2) The age discrepancy between the internal (Guangzhou) and external (Guilin) cohorts might be due to the difference in the treatment preferences of local hospitals or in large cities for young versus old patients. As in endemic areas, the incidence of NPC increases during adolescence and peaks at middle age, and incidence in males is approximately 2- to 3-fold higher than that in females¹. A middle-aged man is usually the primary breadwinner for his family, so NPC in males means a loss of major income and a substantial increase in medical expenses, which imposes a heavy economic burden on the family, especially when the prognosis is poor. Therefore, young patients probably prefer to seek treatment in large cities, where they believe they would receive better medical care, helping them return to society and their life more smoothly. In contrast, elderly patients may choose to seek treatment in local hospitals, which is convenient for their children to care for them. As a result, the age distribution in the internal (Guangzhou) and external (Guilin) cohorts is different.

(3) As the reviewer suggests, we have performed χ^2 test to assess the association of risk groups characterized by the lncRNA signature and the clinical parameters (**Table 2**). *P* values have been added to Table 1 of the revised manuscript.

Table 2. Patient characteristics in the training, internal and external validation cohorts stratified according to the immune-related lncRNA signature

	Training cohort (n=177)				Guangzhou Internal validation cohort (n=177)				Guilin External validation cohort (n=150)			
	No. of Patients	Low risk No. (%)	High risk No. (%)	P	No. of Patients	Low risk No. (%)	High risk No. (%)	P	No. of Patients	Low risk No. (%)	High risk No. (%)	P
Age				0.332				0.323				0.999
<45 years	102	76 (60.3)	26 (51.0)		94	76 (55.5)	18 (45.0)		50	39 (33.1)	11 (34.4)	
≥45 years	75	50 (39.7)	25 (49.0)		83	61 (44.5)	22 (55.0)		100	79 (66.9)	21 (65.6)	
Sex				0.948				0.684				0.590
Male	140	99 (78.6)	41 (80.4)		126	96 (70.1)	30 (75.0)		102	82 (69.5)	20 (62.5)	
Female	37	27 (21.4)	10 (19.6)		51	41 (29.9)	10 (25.0)		48	36 (30.5)	12 (37.5)	
T stage				0.732				0.806				0.197
T1	5	4 (3.2)	1 (2.0)		8	7 (5.1)	1 (2.5)		3	3 (2.5)	0 (0)	
T2	16	10 (7.9)	6 (11.8)		10	7 (5.1)	3 (7.5)		25	21 (17.8)	4 (12.5)	
T3	116	85 (67.5)	31 (60.8)		110	84 (61.3)	26 (65.0)		74	61 (51.7)	13 (40.6)	
T4	40	27 (21.4)	13 (25.5)		49	39 (28.5)	10 (25.0)		48	33 (28.0)	15 (46.9)	
N stage				0.086				0.986				0.514
N0	16	13 (10.3)	3 (5.9)		11	9 (6.6)	2 (5.0)		10	9 (7.6)	1 (3.1)	
N1	76	60 (47.6)	16 (31.4)		78	60 (43.8)	18 (45.0)		54	41 (34.7)	13 (40.6)	
N2	51	33 (26.2)	18 (35.3)		58	45 (32.8)	13 (32.5)		66	54 (45.8)	12 (37.5)	
N3	34	20 (15.9)	14 (27.5)		30	23 (16.8)	7 (17.5)		20	14 (11.9)	6 (18.8)	
EBV DNA (copies/mL)				0.450				0.973				
<2000	79	59 (46.8)	20 (39.2)		69	54 (39.4)	15 (37.5)		/	/	/	
≥2000	98	67 (53.2)	31 (60.8)		108	83 (60.6)	25 (62.5)		/	/	/	
Distant Metastasis				1.6e-06				5.5e-07				5.8e-06
Yes	30	10 (7.9)	20 (39.2)		28	11 (8.0)	17 (42.5)		30	14 (11.9)	16 (50.0)	
No	147	116 (92.1)	31 (60.8)		149	126 (92.0)	23 (57.5)		120	104 (88.1)	16 (50.0)	
Disease progression				8.9e-03				1.1e-05				9.2e-03
Yes	50	28 (22.2)	22 (43.1)		44	23 (16.8)	21 (52.5)		57	38 (32.2)	19 (59.4)	
No	127	98 (77.8)	29 (56.9)		133	114 (83.2)	19 (47.5)		93	80 (67.8)	13 (40.6)	
Death				1.7e-04				3.6e-06				7.0e-04
Yes	36	16 (12.7)	20 (39.2)		33	15 (10.9)	18 (45.0)		31	17 (14.4)	14 (43.8)	
No	141	110 (87.3)	31 (60.8)		144	122 (89.1)	22 (55.0)		119	101 (85.6)	18 (56.2)	

(4) For model construction, we screened lncRNAs associated with distant metastasis-free survival (DMFS) with univariate Cox analysis before using the least absolute shrinkage and selection operator (LASSO) method. Thus, each of the nine lncRNAs was found to be significantly associated with DMFS. In light of the reviewer's comment, we performed χ^2 test to assess the association of each lncRNA and other clinical parameters in all patients. The results of the association analysis are presented in **Table 3** along with the *P* values. The cutoff value for each lncRNA was the threshold that produced the maximum sum of sensitivity and specificity in the ROC curve. As a result, lnc-CDK1-1 was significantly associated with N stage and lnc-STX6-2 was significantly associated with T stage.

Table 3. Association between clinical parameters and individual lncRNA in the signature (presented with *P* values).

	Age	Sex	T stage	N stage	EBV DNA
lnc-TRAPPC6B-2	0.891	0.463	0.821	0.345	0.093
lnc-DRD5-10	0.550	0.999	0.171	0.403	0.111
NR2F2-AS1	0.170	0.303	0.198	0.175	0.199
lnc-CETP-1	0.316	0.073	0.988	0.416	0.803
lnc-CDK1-1	0.691	0.131	0.468	0.004	0.073
LINC02065	0.860	0.147	0.196	0.256	0.228
lnc-POTEH-7	0.391	0.309	0.768	0.637	0.790
lnc-STX6-2	0.902	0.100	0.025	0.138	0.265
lnc-C11orf91-2	0.474	0.696	0.453	0.556	0.920

Reference:

1. Jia, W. H. & Qin, H. D. Non-viral environmental risk factors for nasopharyngeal carcinoma: a systematic review. *Semin Cancer Biol* **22**, 117-126 (2012).
2. Has the selected differentially expressed (DE) lncRNAs been validated by qPCR in the discovery cohort? What is the analysis power to identify DE lncRNAs in the discovery cohort?

Answer:

Thank you very much for the questions.

- (1) In our study, the selected differentially expressed (DE) lncRNAs were not subjected to qRT-PCR in the discovery cohort (n=56, 38 NPC and 18 healthy controls). Instead, similar to many other biomarker studies, we validated and further screened NPC metastasis-related lncRNAs with qRT-PCR analysis in the training cohort (n=177). According to multicohort biomarker studies, researchers often use high-throughput assays to screen DE profiles in small subgroups of patients and then validate the findings with low-throughput methods in larger populations^{2,3}. We expected that it would be more reasonable to validate the DE lncRNAs in the training cohort, which reduces the chance of false positive and false negative discoveries due to a larger sample size. Taking all these factors into account, we performed the validation of DE lncRNAs in the training cohort instead of the discovery cohort.
- (2) The differentially expressed lncRNAs between LA-NPC and healthy controls, and between LA-NPC patients with metastasis and those without metastasis, were identified with cutoffs of $P < 0.05$ and fold change > 1.5 , as described in the Methods section of our manuscript (Page

15, paragraph 2).

References:

2. Nault, J. C. et al. A hepatocellular carcinoma 5-gene score associated with survival of patients after liver resection. *Gastroenterology* **145**, 176-187 (2013).
3. Zhang, J. X. et al. Prognostic and predictive value of a microRNA signature in stage II colon cancer: a microRNA expression analysis. *Lancet Oncol* **14**, 1295-1306 (2013).

3. Are these nine lncRNAs expressed in the cancer cells? It is worthwhile to identify the source of lncRNA expression detected in this study.

Answer:

Thank you for your valuable comment. Based on the microarray analysis results, we believe the nine lncRNAs are not only expressed in cancer cells, as they were also detected in normal nasopharyngeal tissue samples. To further confirm this inference, we tried to isolate different types of cells from fresh NPC tissues to explore the source of the nine lncRNAs.

We collected NPC biopsies from the primary tumor site by endoscopy. After full digestion and antibody incubation, we isolated tumor cells and immune cells with fluorescence-activated cell sorting (FACS) using a flow cytometer (MoFlo Astrios, Beckman-Coulter). APC anti-human CD45 (Cat# 304011, Biolegend) and PE anti-human EpCAM (Cat# 12-9326-42, eBioscience) were used to identify tumor cells (EpCAM⁺CD45⁻) and immune cells (EpCAM⁻CD45⁺). TRIzol reagent (Invitrogen) was used to isolate total RNA. PCR was performed using SYBR Green qPCR SuperMix-UDG reagents (Invitrogen) on a LightCycler 480 detection system (Roche) with β -actin as an endogenous control. We collected three samples for qRT-PCR analysis, and all lncRNAs were detected in both cancer cells and immune cells (**Figure 1**).

It has been reported that some lncRNAs play essential roles in tumor progression and lead to considerable differences in clinical outcomes, which have been found to be expressed in malignant cells and nonmalignant cells. For instance, in breast cancer, lncRNA NKILA can regulate T cell sensitivity to activation-induced cell death in T lymphocytes⁴, and it can suppress TGF- β -induced epithelial-mesenchymal transition of cancer cells by blocking NF- κ B signaling⁵. More importantly, lncRNA dysregulation is not an isolated event limited to lncRNA-expressing cells. Conversely, it would have an extensive impact on the tumor microenvironment (TME). For example, lncRNA

LIMIT from cancer cells can promote the T-cell-mediated tumor immune response and enhance immunotherapy efficacy⁶. LncRNA HISLA from the tumor-associated macrophages can regulate aerobic glycolysis of breast cancer cells through extracellular vesicles⁷. These findings suggest subtle regulation and complicated interactions between different cell types in the TME. Likewise, dysregulation of the nine lncRNAs probably reflects changes in the TME under premetastatic conditions, as they are expressed in both the tumor and immune cells and have been associated with pathways related to metastasis and immune function.

Figure 1. Expression of the nine lncRNA in isolated cancer and immune cells from three NPC biopsy tissues. Error bars represent standard deviation (some standard deviations are too small to see in the figure).

References:

- Huang, D. et al. NKILA lncRNA promotes tumor immune evasion by sensitizing T cells to activation-induced cell death. *Nat Immunol* **19**, 1112-1125 (2018).
- Wu, W. et al. LncRNA NKILA suppresses TGF- β -induced epithelial-mesenchymal transition by blocking NF- κ B signaling in breast cancer. *Int J Cancer* **143**, 2213-2224 (2018).
- Li, G. et al. LIMIT is an immunogenic lncRNA in cancer immunity and immunotherapy. *Nat Cell Biol* **23**, 526-537 (2021).
- Chen, F. et al. Extracellular vesicle-packaged HIF-1 α -stabilizing lncRNA from tumour-associated macrophages regulates aerobic glycolysis of breast cancer cells. *Nat Cell Biol* **21**, 498-510 (2019).

4. In Figure 6, the difference between the high- and low-risk groups was observed in the B cells and CD8+ T cell infiltration. Can these two cell populations be the predictors for NPC metastasis as well?

Answer:

Thank you very much for this question. As the reviewer suggests, we investigated the prognostic value of B cells and CD8⁺ T cells in our cohort (n=99). The results showed that the intratumoral and stromal CD8⁺ T cell infiltration were significantly lower in the metastatic group than those in the nonmetastatic group (**Figure 2A-B**). The Kaplan–Meier plots also displayed significantly different DMFS in patients with high and low CD8⁺ T cell infiltration (**Figure 2C-D**). Meanwhile, we found that the intratumoral and stromal CD20⁺ B cell infiltration had no significant difference between the metastatic and nonmetastatic groups (**Figure 2E-F**). Also, high and low level of CD20⁺ B cell infiltration were not predictors for DMFS (**Figure 2G-H**).

In our study, CD8⁺ T cell infiltration but not CD20⁺ B cells was predictor of metastasis, but we also acknowledge that the results might be limited by the relatively small sample size (n=99) and should be further validated in a larger cohort before drawing conclusions.

Figure 2. Association of B cells and CD8⁺ T cells with metastasis in NPC (n=99). (A-B) Comparison of intratumoral (A) and stromal (B) CD8⁺ T cells in patients in metastatic and non-metastatic group. The comparison was performed with Student's *t*-test. (C-D) Kaplan-Meier curves for distant metastasis-free survival (DMFS) of patients according to the infiltration of intratumoral (C) and stromal (D) CD8⁺ T cells. *P* values were calculated by log-rank test. (E-F) Comparison of intratumoral (E) and stromal (F) CD20⁺ B cells in patients in metastatic and non-metastatic group. The comparison was performed with Student's *t*-test. In each box plot, the centerline represents the median, the bounds represent the 1st and 3rd quartiles and whiskers extends from the hinge to the largest value no further than 1.5 * interquartile range (IQR) from the hinge. Each dot represents data point for individual patients. (G-H) Kaplan-Meier curves for DMFS of patients according to the infiltration of intratumoral (G) and stromal (H) CD20⁺ B cells. *P* values were calculated by log-rank test.

5. It is unclear whether the prediction accuracy can be further boosted by combining three variables (lncRNA signature, pre-EBV and N stage) together in Guangzhou training and validation cohorts. Both cohorts have the pre-EBV data.

Answer:

We appreciate your advice. Using the same method described in the manuscript, we constructed a model by combining three variables (lncRNA signature, pretreatment EBV DNA and N stage) using multivariate Cox analysis in the training cohort (**Table 4**), followed by validation in the internal cohort. The combined three-variable model also improved the efficiency of metastasis prediction compared with pretreatment EBV DNA or N stage alone (**Figure 3**), which was confirmed in the internal validation cohort. We also noted that the accuracy of the three-variable model was not inferior compared to that of the two-variable model (combining the nine-lncRNA signature and N stage) (**Figure 3**).

Table 4. Summary of the multivariable analysis of prognostic factors for distant metastasis-free survival and risk weight in the Guangzhou training cohort.

Variables	β Coefficient	HR	95%CI for HR	P	Weight
lncRNA signature					
Low risk	1	1			
High risk	1.63	5.10	2.36-11.01	3.46e-5	2
N stage					
N0-1	1	1			
N2-3	0.88	2.41	1.04-5.56	0.039	1
EBV DNA					
≤ 2000 copies/ml	1	1			
> 2000 copies/ml	1.05	2.86	1.15-7.08	0.024	1

Figure 3. Performance of the three-variable model. ROC curve analysis of the three-variable model, the lncRNA signature, N stage and pretreatment EBV DNA for predicting distant metastasis in patients in the training (A, n=177) and internal validation (B, n=177) cohorts. ROC curve analysis of the three-variable and lncRNA-N stage model for predicting distant metastasis in patients in the training (C) and internal validation (D) cohorts. DeLong's test was used to estimate the *P* values.

6. The Cox proportional hazards model was used in the analysis. How to evaluate the assumption of the proportional hazard for the Cox model is valid in the analysis?

Answer:

Thank you for your question. The nine-lncRNA signature is the focus of the current study. Using Cox analysis, we found that it was significantly associated with DMFS. The Kaplan–Meier plots showed no cross-section between survival lines of the high- and low-risk groups, which primarily indicated that the proportional hazard (PH) assumption is likely to hold⁸. To further assess proportionality, we tested the PH assumption by Schoenfeld residuals⁹ and test time and covariate interaction for statistical significance⁸ using the "survival" and "survminer" packages in R software (version 4.0.3). The Schoenfeld residual test (**Figure 4**) and the interaction of the lncRNA signature and time (**Table 5**) were not statistically significant.

Other covariates examined include host factors (i.e., age and sex), tumor factors (i.e., T and N stage) and pretreatment EBV DNA using the above methods, as well as a global test for all

covariates after fitting the Cox regression model. As a result, the PH assumption is also evidenced by Schoenfeld residual plots (**Figure 5**) and a global test for all covariates after fitting the Cox regression model (Guangzhou training cohort: $P_{\text{global}} = 0.87$; Guangzhou internal validation cohort: $P_{\text{global}} = 0.50$; Guilin external validation cohort: $P_{\text{global}} = 0.19$). The results of the time and covariate interaction also support the PH assumption (**Table 5**).

Figure 4. The Schoenfeld residual plots of lncRNA signature in three cohorts, including (A) Guangzhou training cohort, (B) Guangzhou internal validation cohort and (C) Guilin external validation cohort.

Table 5. Statistical results of time and covariates interaction (presented with P value).

	Guangzhou training cohort	Guangzhou internal cohort	Guilin external cohort
lncRNA signature	0.60	0.14	0.62
Age	0.12	0.39	0.11
Sex	0.38	0.82	0.65
T stage	0.09	0.70	0.53
N stage	0.63	0.31	0.14
EBV DNA	0.55	0.53	NA

Figure 5. The Schoenfeld residual plots of covariates in three cohorts. Plots show (A) age, (B) sex, (C) T stage, (D) N stage and (E) pretreatment EBV DNA in Guangzhou training cohort; (F) age, (G) sex, (H) T stage, (I) N stage and (J) pretreatment EBV DNA in Guangzhou internal validation cohort; (K) age, (L) sex, (M) T stage and (N) N stage in Guilin external validation cohort. P values were estimated by Schoenfeld residual test.

References:

- Kuitunen, I., Ponkilainen, V. T., Uimonen, M. M., Eskelinen, A. & Reito, A. Testing the proportional hazards assumption in cox regression and dealing with possible non-proportionality in total joint arthroplasty research: methodological perspectives and review. *BMC Musculoskeletal Disord* **22**, 489 (2021).
- Schoenfeld, D. Partial residuals for the proportional hazards regression model. 239-241.

7. What is the possible reason for upregulation/downregulation of these lncRNA expression in NPC tissues?

Answer:

Thank you very much for the question. As reported, the discrepant expression of lncRNAs in cancer can be regulated at the genome level, transcription level, and posttranscription level.

At the genome level, loss of heterozygosity (LOH) has been found in various cancers, which results in the depletion of tumor-suppressing genes, including lncRNAs¹⁰. The downregulated expression of lncRNAs has become a critical step in elucidating the molecular pathogenesis of tumors. Additionally, somatic copy-number alterations (SCNAs) of lncRNAs are common in most solid cancer genomes. The expression levels of these lncRNAs are closely correlated with their copy number abnormalities¹¹. In NPC, some noncoding genes were identified within the frequent deletion or duplication regions and contributed to cancer genetic susceptibility¹². For example, lncRNA MEG3 was inactive because of its location in a common copy number deletion region, chromosome 14q32¹³. These findings suggest that the chromatin abnormalities may be an essential mechanism involved in the lncRNA dysregulation.

Similar to protein-coding gene, lncRNA is regulated by typical epigenetic and transcriptional mechanisms. For instance, the activating marker (H3K27ac) is strongly enriched in the promoter region of lncRNA CCAT1, which activates CCAT1 transcription and upregulates its expression¹⁴. A similar phenomenon can be found in the AFAP1-AS1 promoter under trastuzumab treatment, which confers chemoresistance to the cancer cells¹⁵. Additionally, the methylation of CpG islands inhibits the binding of DNA to transcription-associated proteins and mediates gene silencing¹⁶. Therefore, the DNA methylation status of a gene promoter region is negatively correlated with its expression^{17,18}, which contributes to the carcinogenesis and tumor progression. It is worth noting that aberrant DNA methylation was found to be a frequent event in NPC, and hypermethylation status was found in some specific genetic loci, indicating that some tumor-suppressing lncRNAs may be silenced by promoter hypermethylation in NPC^{19,20}.

Moreover, lncRNA expression can also be regulated at the posttranscriptional level. One of the key features of this mechanism is mediated by RNA binding proteins (RBPs), which can alter RNA stability by controlling the molecular environment²¹⁻²³. For instance, IGF2BP1 accelerates

the catabolism of lncRNA HULC and results its downregulation in liver cancer²⁴. HuR stabilizes HGBC and maintains its high expression in gallbladder cancer²⁵. In NPC, IGF2BP2 binds with TINCR to slow down its RNA degradation and upregulates TINCR's expression²⁶.

No studies have yet explored the potential reasons for the dysregulated expression of the nine lncRNAs in the immune-related signature in NPC, which entails a significant amount of research. Upcoming research by our group will hopefully bring more clarity to these issues.

References:

10. Zhang, J. et al. Long noncoding RNA TSLNC8 is a tumor suppressor that inactivates the interleukin-6/STAT3 signaling pathway. *Hepatology* **67**, 171-187 (2018).
11. Hu, X. et al. A functional genomic approach identifies FAL1 as an oncogenic long noncoding RNA that associates with BMI1 and represses p21 expression in cancer. *Cancer Cell* **26**, 344-357 (2014).
12. Tse, K. P. et al. A gender-specific association of CNV at 6p21.3 with NPC susceptibility. *Hum Mol Genet* **20**, 2889-2896 (2011).
13. Chak, W. P. et al. Downregulation of long non-coding RNA MEG3 in nasopharyngeal carcinoma. *Mol Carcinog* **56**, 1041-1054 (2017).
14. Zhang, E. et al. H3K27 acetylation activated-long non-coding RNA CCAT1 affects cell proliferation and migration by regulating SPRY4 and HOXB13 expression in esophageal squamous cell carcinoma. *Nucleic Acids Res* **45**, 3086-3101 (2017).
15. Han, M. et al. Exosome-mediated lncRNA AFAP1-AS1 promotes trastuzumab resistance through binding with AUF1 and activating ERBB2 translation. *Mol Cancer* **19**, 26 (2020).
16. Jones, P. A. Functions of DNA methylation: islands, start sites, gene bodies and beyond. *Nat Rev Genet* **13**, 484-492 (2012).
17. He, Y. et al. DNMT1-mediated lncRNA MEG3 methylation accelerates endothelial-mesenchymal transition in diabetic retinopathy through the PI3K/Akt/mTOR signaling pathway. *Am J Physiol Endocrinol Metab* **320**, E598-E608 (2021).
18. Li, R. et al. Methylation and transcriptome analysis reveal lung adenocarcinoma-specific diagnostic biomarkers. *J Transl Med* **17**, 324 (2019).
19. Dai, W. et al. Comparative methylome analysis in solid tumors reveals aberrant methylation at chromosome 6p in nasopharyngeal carcinoma. *Cancer Med* **4**, 1079-1090 (2015).
20. Lung, H. L. et al. Deciphering the molecular genetic basis of NPC through functional approaches. *Semin Cancer Biol* **22**, 87-95 (2012).
21. Nyati, K. K., Zaman, M. M., Sharma, P. & Kishimoto, T. Arid5a, an RNA-Binding Protein in Immune Regulation: RNA Stability, Inflammation, and Autoimmunity. *Trends Immunol* **41**, 255-268 (2020).
22. Ray, D. et al. A compendium of RNA-binding motifs for decoding gene regulation. *Nature* **499**, 172-177 (2013).
23. Qin, H. et al. RNA-binding proteins in tumor progression. *J Hematol Oncol* **13**, 90 (2020).
24. Hammerle, M. et al. Posttranscriptional destabilization of the liver-specific long noncoding RNA HULC by the IGF2 mRNA-binding protein 1 (IGF2BP1). *Hepatology* **58**, 1703-1712 (2013).
25. Hu, Y. P. et al. LncRNA-HGBC stabilized by HuR promotes gallbladder cancer progression by

- regulating miR-502-3p/SET/AKT axis. *Mol Cancer* **18**, 167 (2019).
26. Zheng, Z. Q. et al. Long Noncoding RNA TINCR-Mediated Regulation of Acetyl-CoA Metabolism Promotes Nasopharyngeal Carcinoma Progression and Chemoresistance. *Cancer Res* **80**, 5174-5188 (2020).

8. A digital pathology method was used in the IHC analyses in this study. The details about how to verify the results from the digital analysis by HALO software should be added in the methods.

Answer:

Thank you for your helpful suggestion. For digital quantitation of CD8⁺ T cells and CD20⁺ B cells, the full view of sequential hematoxylin and eosin (H&E) staining and immunohistochemistry (IHC) slides for each sample were digitally scanned using a ZEISS Axio Scan.Z1 microscope at ×200 magnification. Digital image analyses were performed using HALO software (version 3.3.2541.323, Indica Labs, USA)^{27,28}. For each slide, tumor nest and stromal areas were manually annotated by two experienced pathologists who were blinded to the clinical data. HE-, CD8- and CD20-stained serial tissue sections were registered and aligned using the "Landmarks" function in HALO²⁹. In this way, manual annotations of the tumor and stromal areas on the H&E slides were automatically transferred to aligned sections, annotating the entire tumor and stromal areas in the sequential IHC slides. The areas for analysis were carefully selected not to contain folds. Then, we established training datasets for CD8- or CD20-stained images by randomly selecting 30 slides and performed nuclei segmentation and cell classification by the HALO multiplex IHC algorithm (version 3.1.4)^{28,30}. Automated classification of the positive staining cells was performed based on the cytonuclear features such as stain intensity, size and roundness, the thresholds of which were trained by pathologists²⁷. Pathologists validated the good performance of the trained algorithm in identifying positive cells through visual inspection³¹, which was then applied to the rest of the slides using the same parameters. Finally, the algorithm automatically quantified the numbers of positive cells and the areas of tumor and stroma across the whole slide image. For each patient, the positively stained cell density was presented as the cell count per mm².

To address the reviewer's concern, we have added these details about the digital pathology analysis to the Methods section of the revised manuscript (Page 17, paragraph 1).

References:

27. Challoner, B. R. et al. Computational Image Analysis of T-Cell Infiltrates in Resectable Gastric Cancer: Association with Survival and Molecular Subtypes. *J Natl Cancer Inst* **113**, 88-98 (2021).
28. Hofman, P. et al. Multiplexed Immunohistochemistry for Molecular and Immune Profiling in Lung Cancer-Just About Ready for Prime-Time? *Cancers (Basel)* **11** (2019).
29. Koelzer, V. H., Sirinukunwattana, K., Rittscher, J. & Mertz, K. D. Precision immunoprofiling by image analysis and artificial intelligence. *Virchows Arch* **474**, 511-522 (2019).
30. Koelzer, V. H. et al. Digital image analysis improves precision of PD-L1 scoring in cutaneous melanoma. *Histopathology* **73**, 397-406 (2018).
31. Flaifel, A. et al. PD-L1 Expression and Clinical Outcomes to Cabozantinib, Everolimus, and Sunitinib in Patients with Metastatic Renal Cell Carcinoma: Analysis of the Randomized Clinical Trials METEOR and CABOSUN. *Clin Cancer Res* **25**, 6080-6088 (2019).

Minor comments

1. The font size for the pathways in Figure 5C is too small.
2. The circo plot in Figure 5C is difficult to read. What does the size of pathways 1-22 mean in the plot? Why the size of the pathway 3 "Angiogenesis" is much smaller than the pathway 1 "EMT"? More details for the figure legend should be added.

Answers:

Thank you for your valuable comments. The size of the pathway represents the number of genes in the gene set of that pathway. For example, there were 36 genes in the "angiogenesis" gene set, which is much smaller than other gene sets such as "EMT" (200 genes). Therefore, the size of the "angiogenesis" pathway is relatively smaller.

To address the reviewer's concern, we have modified the figure and related legend of Figure 5C in the revised manuscript to make it clearer for the readers.

(Manuscript) Figure 5C. Circos plot showing the significantly enriched pathways between the high- and low- risk groups. The size of each sector represents the number of genes in the labeled pathway. The color of the outmost circle and inner bar plots represent the category of pathways. The size of the second outer circle represents the percentage of genes contributing to the enrichment score, and its color represents the magnitude of the statistical significance (shown as $-\log_{10}(P \text{ value})$). The color of the third circle indicates that the labeled pathway is upregulated in the high- or low-risk group. The size of the inner bar plot shows the normalized enrichment score.

Response to Reviewer #2:

A lncRNA signature associated with tumour immune heterogeneity predicts distant metastasis in locoregionally advanced nasopharyngeal carcinoma

The authors have identified a set of 9 long-non-coding RNAs to be predictive of locoregionally advanced nasopharyngeal carcinoma. The authors derive a weighted formula based on the expression levels (by qRT-PCR) of these 9 lncRNAs and show that it is predictive of LA-NPC metastasis in two cohorts of patients. The authors also show that the 9 lncRNAs are independently prognostic relative to other known prognostic factors and correlate with the level of immune infiltration.

Comments:

1. Introduction: The language needs overall work (academic style writing).

Answer:

Thank you for your valuable comment. We have modified the manuscript carefully and have further used English language editing services to provide language editing assistance. The major modified parts are indicated in the revised manuscript. The editing certificate has been attached as follows.

[REDACTED]

2. Line 198-215: Does the derived risk score weights/formula sensitive to the method/kit used for qRT-PCR? If the authors propose this model to be used by other investigators, it is important to test its sensitivity outside the developed lab.

Answer:

Thank you very much for the question. Although many commercially available qRT-PCR kits and platforms generally perform the same function, the quantification results differ and maybe affected by the reagents, hardware and software design used^{1,2}. For instance, the detection of EBV DNA, an indicator for NPC disease monitoring, can yield large variability using quantitative PCR assays without harmonization even in experienced clinical labs³. A tremendous amount of work has been done to reduce this variability, including the use of common calibrators and a PCR master mix. Researchers from Stanford University and another four centers in Hong Kong and Taiwan reported the standardization of quantitative PCR to reduce the variability of EBV DNA quantification across different clinical laboratories in various hospitals/countries³. To reduce the variability, the detection reagents, PCR primer for EBV DNA and detection protocol should be the same as those credentialed by Stanford University.

Likewise, in terms of our proposed lncRNA signature, standardization should be established. When this signature model used by other investigators, the PCR reagents, protocol and risk calculation formula should remain uniform to reduce variability.

References:

1. Lu, S., Smith, A. P., Moore, D. & Lee, N. M. Different real-time PCR systems yield different gene expression values. *Mol Cell Probes* **24**, 315-320 (2010).
2. Yang, J. et al. The source of SYBR green master mix determines outcome of nucleic acid amplification reactions. *BMC Res Notes* **9**, 292 (2016).
3. Le, Q. T. et al. An international collaboration to harmonize the quantitative plasma Epstein-Barr virus DNA assay for future biomarker-guided trials in nasopharyngeal carcinoma. *Clin Cancer Res* **19**, 2208-2215 (2013).

Line 261: Please add more details of this analysis in the methods section. It is not clear how the correlation was performed, cut-off used, and so on.

Answer:

Thank you very much for the valuable comments. In this study, we investigated the functional

contexts of the nine lncRNAs in our signature based on the guilt-by-association principle, which uses correlation information to predict new gene members in the functional call of genes⁴. This approach is based on correlation analysis between matching lncRNA and protein-coding mRNA expression in combination with enrichment strategies. It contains the following three steps: (i) for each lncRNA, an expression-correlation matrix (Spearman's rank) was built by combining lncRNA and matching mRNA expression data; (ii) the expression-correlation matrix was ranked by correlation coefficient in descending order; and (iii) the sorted expression-correlation matrix was used as input for gene set enrichment analysis (GSEA) (hallmark gene set list obtained from MSigDb and immune-related pathways from the ImmPort website⁵). Pathways with $P < 0.05$, $FDR < 0.25$ and absolute normalized enrichment score ($|NES| > 1$) were considered significant and were used to infer the functions of the corresponding lncRNAs.

In light of the reviewer's comments, we have added these details about functional enrichment analysis to the Methods section of the revised manuscript (Page 16, paragraph 2).

References:

4. Lefever, S. et al. decodeRNA- predicting non-coding RNA functions using guilt-by-association. *Database (Oxford)* **2017** (2017).
5. Bhattacharya, S. et al. ImmPort, toward repurposing of open access immunological assay data for translational and clinical research. *Sci Data* **5**, 180015 (2018).

4. Line 292: Can the authors explain this analysis further in methods?

- Was that tissue considered as a whole or only tumor-specific regions were used in the analysis?
- Was the total size of the tissue taken into consideration and normalized prior to showing a significant difference in immune T/B cells between patients?

Answer:

Thank you very much for your questions. The histopathologic results presented in our study were obtained by measuring the entire tumor tissue in the whole sections. To annotate intratumoral and stromal areas, we referred to the criteria of intratumoral and stromal tumor-infiltrating lymphocyte (TIL) evaluation described by the International Immuno-Oncology Biomarker Working Group⁶⁻⁸. According to the criteria, we defined the intratumoral CD8⁺ T cells and CD20⁺ B cells as those in tumor nests of the whole tissue sections having cell-to-cell contact with no intervening stroma and directly interacting with cancer cells, while stromal CD8⁺ T cells and CD20⁺ B cells are dispersed

in the stroma between the tumor nests and do not directly contact cancer cells. In addition, zones with crush artifacts, necrosis and fibrosis were excluded from the analysis. The same standard has been used in many studies for immune infiltration evaluation in various cancers⁹⁻¹², including our previous large-scale cohort study in NPC¹³.

We divided positive cell counts by the corresponding areas when calculating the density of CD8⁺ T cells and CD20⁺ B cells. More specifically, the density of intratumoral CD8⁺ T cells was equal to intratumoral CD8⁺ positive cell counts divided by intratumoral areas (mm²), which were both derived from HALO software. This method is widely used in digital image analysis^{14,15} and considered to be useful for standardization, since it determines the number of infiltrating cells as an exact measurement contrary to the approximate semiquantitative evaluation⁶. In this way, the effect of tumor size on immune infiltration assessment has been taken into account.

To address the reviewer's concern, we have added more details about the digital pathology analysis to the Methods section of the revised manuscript (Page 17, paragraph 1) and modified the pipeline of digital pathology analysis in Fig. 6B.

B
(Manuscript) Fig. 6B Digital pathology analysis pipeline. For each sample, H&E and IHC slides were scanned, registered and aligned in HALO software. Manual annotation of tumor nest and stroma areas on the H&E image were automatically synchronized to IHC image. Positively stained cells in IHC image were identified by the Multiplex IHC algorithm. Quantification of positively stained cells and the areas of tumor nest and stroma were automatically generated by HALO software. The scale bar represents 25 μm .

References:

6. Salgado, R. et al. The evaluation of tumor-infiltrating lymphocytes (TILs) in breast cancer: recommendations by an International TILs Working Group 2014. *Ann Oncol* **26**, 259-271 (2015).
7. Hendry, S. et al. Assessing Tumor-infiltrating Lymphocytes in Solid Tumors: A Practical Review for Pathologists and Proposal for a Standardized Method From the International Immunooncology Biomarkers Working Group: Part 1: Assessing the Host Immune Response, TILs in Invasive Breast Carcinoma and Ductal Carcinoma In Situ, Metastatic Tumor Deposits and Areas for Further Research. *Adv Anat Pathol* **24**, 235-251 (2017).
8. Hendry, S. et al. Assessing Tumor-Infiltrating Lymphocytes in Solid Tumors: A Practical Review for Pathologists and Proposal for a Standardized Method from the International Immuno-Oncology Biomarkers Working Group: Part 2: TILs in Melanoma, Gastrointestinal Tract Carcinomas, Non-Small Cell Lung Carcinoma and Mesothelioma, Endometrial and Ovarian Carcinomas, Squamous Cell Carcinoma of the Head and Neck, Genitourinary Carcinomas, and Primary Brain Tumors. *Adv Anat Pathol* **24**, 311-335 (2017).
9. Brambilla, E. et al. Prognostic Effect of Tumor Lymphocytic Infiltration in Resectable Non-Small-Cell Lung Cancer. *J Clin Oncol* **34**, 1223-1230 (2016).
10. Loi, S. et al. Tumor-Infiltrating Lymphocytes and Prognosis: A Pooled Individual Patient Analysis of Early-Stage Triple-Negative Breast Cancers. *J Clin Oncol* **37**, 559-569 (2019).
11. Desmedt, C. et al. Immune Infiltration in Invasive Lobular Breast Cancer. *J Natl Cancer Inst* **110**, 768-776 (2018).
12. Sanz-Pamplona, R. et al. Lymphocytic infiltration in stage II microsatellite stable colorectal tumors: A retrospective prognosis biomarker analysis. *PLoS Med* **17**, e1003292 (2020).
13. Wang, Y. Q. et al. Prognostic significance of tumor-infiltrating lymphocytes in nondisseminated nasopharyngeal carcinoma: A large-scale cohort study. *Int J Cancer* **142**, 2558-2566 (2018).
14. Nearchou, I. P. et al. Automated Analysis of Lymphocytic Infiltration, Tumor Budding, and Their Spatial Relationship Improves Prognostic Accuracy in Colorectal Cancer. *Cancer Immunol Res* **7**, 609-620 (2019).
15. Abe, N. et al. Quantitative digital image analysis of tumor-infiltrating lymphocytes in HER2-positive breast cancer. *Virchows Arch* **476**, 701-709 (2020).

Minor:

5. Line 83: Please provide a reference.

Answer:

Thank you for the helpful suggestion. After carefully going through this sentence, we found that several words were mistakenly deleted, which changed our intended meaning. The sentence should be "It is generally recognized that lncRNAs are a group of transcripts that are exquisitely regulated and are more cell-type specific to a greater degree than mRNA"¹⁶.

To address the reviewer's concern, we have corrected the sentence in the Introduction section of the revised manuscript and provided supporting reference (Page 4, paragraph 2).

Reference:

16. Cabili, M. N. et al. Integrative annotation of human large intergenic noncoding RNAs reveals global properties and specific subclasses. *Genes Dev* **25**, 1915-1927 (2011).

6. Line 87: The previous lines (as written) do not "demonstrate" lncRNAs as promising biomarkers. Just rewording to be suggestive with improving the overall flow.

Answer:

Thank you for your valuable suggestion. As the reviewer's suggestion, we have reworded the sentences "A large number of lncRNAs have been reported to facilitate tumor growth, migration and invasion^{17,18}, and to modulate immune response¹⁹ and signaling pathways²⁰, thus contributing to distant metastasis²¹. Consequently, prognostic lncRNA signatures have been developed, and have shown promising accuracy in predicting tumor metastasis²²⁻²⁴. However, studies have yet reported whether the lncRNAs could serve as metastatic predictors for NPC patients." in the Introduction Section of our revised manuscript (Page 4, paragraph 2).

References:

17. Yang, R. et al. Long noncoding RNA PVT1 promotes tumor growth and predicts poor prognosis in patients with diffuse large B-cell lymphoma. *Cancer Commun (Lond)* **40**, 551-555 (2020).
18. Xie, J.-J. et al. Super-Enhancer-Driven Long Non-Coding RNA LINC01503, Regulated by TP63, Is Over-Expressed and Oncogenic in Squamous Cell Carcinoma. *Gastroenterology* **154**, 2137-2151.e2131 (2018).
19. Huang, D. et al. NKILA lncRNA promotes tumor immune evasion by sensitizing T cells to activation-induced cell death. *Nat Immunol* **19**, 1112-1125 (2018).
20. Li, C. et al. A ROR1-HER3-lncRNA signalling axis modulates the Hippo-YAP pathway to regulate bone metastasis. *Nat Cell Biol* **19**, 106-119 (2017).
21. Liu, S. J., Dang, H. X., Lim, D. A., Feng, F. Y. & Maher, C. A. Long noncoding RNAs in cancer metastasis. *Nat Rev Cancer* (2021).
22. Prensner, J. R. et al. RNA biomarkers associated with metastatic progression in prostate cancer: a multi-institutional high-throughput analysis of SChLAP1. *Lancet Oncol* **15**, 1469-1480 (2014).
23. Gupta, R. A. et al. Long non-coding RNA HOTAIR reprograms chromatin state to promote cancer metastasis. *Nature* **464**, 1071-1076 (2010).
24. Ji, P. et al. MALAT-1, a novel noncoding RNA, and thymosin beta4 predict metastasis and survival in early-stage non-small cell lung cancer. *Oncogene* **22**, 8031-8041 (2003).

7. Minor Typos/misc

Line 82: extra space while citing 8

Line 84: extra space after citing 9-11

Line 87: extra space while citing 16

Line 91: Did the authors mean high throughput?

Line 94: Please check grammar

Line 97: unavailability?

Line 268: Please cite Immport

Answer:

Thank you for your thorough review. We have corrected the minor typos and misnomers you mentioned above. In addition, we have proofread the manuscript and further employed an English language editing service to provide language editing assistance for the revised manuscript.

Response to Reviewer #3:

In this work, the authors integrated multiple NPC data and identified nine lncRNA related to prognosis through a series of conventional bioinformatics methods. The authors developed a nine-lncRNA module which were crucial to the NPC patients' prognosis and metastasis and validated the module in other independent data to prove the accuracy. The work prioritized lncRNA biomarker in NPC and provides a novel prognosis module for NPC subtype. However, there are some issues need to be solved.

Major Point:

1. The description of differential expression analysis is indistinct. The author should add specific methods or the R package. Besides, the author can provide two Supplementary table containing the detail result, such as the p value, FDR value for each gene.

Answer:

Thank you very much for your advice. In the microarray data analysis, the raw data were extracted using the Agilent Feature Extraction software (version 11.0.1.1). Quantile normalization, quality control and differential expression analysis were performed with GeneSpring GX v12.1 software package (Agilent Technologies). After normalization, transcript-specific lncRNAs probes labeled as "present" in more than or equal to half of the samples were considered high-qualified for the biomarker selection. Differential expression analyses were performed based on *t*-test between samples from 18 LA-NPC patients and 18 healthy controls, and between 10 paired samples from LA-NPC developed with or without posttreatment distant metastasis. The lncRNAs showing significant differential expression between the two groups were identified with cutoffs of *P* value < 0.05 and Fold Change > 1.5.

As the reviewer suggests, we have modified the Methods section (Page 15, paragraph 2). We also summarized the results of differential expression analysis with fold change, *P* value and FDR in Supplementary table S3.

2. Although lncRNA is very important in the process of disease development, protein coding genes contain a wider range of genetic information. Why does the author only analyze the prognosis

factor based on lncRNA expression profile, rather than integrate the two expression profiles?

Answer:

Thank you very much for your valuable comments. Although the function of protein-coding genes has been substantially elucidated, emerging technologies are expanding investigators' abilities to annotate lncRNAs, and complex molecular mechanisms for lncRNAs are now being demonstrated. LncRNAs have now been linked with the context of most, if not all, of the classic hallmarks of cancer, including tumor metastasis¹. It has been reported that lncRNAs promote tumor metastasis by facilitating invasion and migration, modifying EMT, regulating metastatic colonization, and altering tumor microenvironment¹. Our previous studies have also reported the roles of lncRNAs involved in NPC metastasis. For instance, DANCR promotes metastasis by interacting with the NF90/NF45 complex², and FAM225A promotes metastasis by acting as a ceRNA to sponge miR-590-3p/miR-1275 and upregulate ITGB3³. These studies motivate our interest to determine whether lncRNAs could serve as metastasis predictors in NPC.

Increasing studies reveal that lncRNAs modulate mRNAs expression through various ways, including controlling mRNA stability, splicing, and translation⁴, which illustrates the complicated relationship between lncRNAs and mRNAs. As a result, a Cox model incorporating the lncRNAs and mRNAs for prognosis might not be suitable to reflect their subtle and complicated interactions. To our knowledge, studies usually integrate mRNA and lncRNA expression profiles to construct a ceRNA network⁵⁻⁷. This network probably helps to illustrate the regulatory mechanisms between those RNAs, but it is rarely used directly to construct a biomarker for clinical practice as it might not be easily generated by a mathematical formula. An applicable model requires a clear definition of predictors and reproducible measurement methods available in clinical practice⁸. Therefore, we believe that a prognostic model calculated with a simple mathematical formula is of clinical value. Considering the above, we constructed a lncRNA-based signature in present study. Subsequently, the proposed lncRNA signature was proven to be a robust prognostic indicator for distinguishing patients with a high risk of posttreatment distant metastasis in NPC.

In light of the reviewer's comment, we will explore an appropriate method for integrating mRNA and lncRNA data and evaluate whether integrating mRNAs with lncRNAs could further enhance the prediction accuracy in our future work.

References:

1. Liu, S. J., Dang, H. X., Lim, D. A., Feng, F. Y. & Maher, C. A. Long noncoding RNAs in cancer metastasis. *Nat Rev Cancer* (2021).
2. Wen, X. et al. Long non-coding RNA DANCR stabilizes HIF-1alpha and promotes metastasis by interacting with NF90/NF45 complex in nasopharyngeal carcinoma. *Theranostics* **8**, 5676-5689 (2018).
3. Zheng, Z. Q. et al. Long Noncoding RNA FAM225A Promotes Nasopharyngeal Carcinoma Tumorigenesis and Metastasis by Acting as ceRNA to Sponge miR-590-3p/miR-1275 and Upregulate ITGB3. *Cancer Res* **79**, 4612-4626 (2019).
4. Schmitt, A. M. & Chang, H. Y. Long Noncoding RNAs in Cancer Pathways. *Cancer Cell* **29**, 452-463 (2016).
5. Cheng, Y. et al. Identification of circRNA-lncRNA-miRNA-mRNA Competitive Endogenous RNA Network as Novel Prognostic Markers for Acute Myeloid Leukemia. *Genes (Basel)* **11** (2020).
6. Wang, L. X. et al. Integrative Analysis of Long Noncoding RNA (lncRNA), microRNA (miRNA) and mRNA Expression and Construction of a Competing Endogenous RNA (ceRNA) Network in Metastatic Melanoma. *Med Sci Monit* **25**, 2896-2907 (2019).
7. Wang, W. et al. A novel mRNA-miRNA-lncRNA competing endogenous RNA triple sub-network associated with prognosis of pancreatic cancer. *Aging (Albany NY)* **11**, 2610-2627 (2019).
8. Moons, K. G., Altman, D. G., Vergouwe, Y. & Royston, P. Prognosis and prognostic research: application and impact of prognostic models in clinical practice. *BMJ* **338**, b606 (2009).

3. In the process of model verification, the author uses the same threshold. This value is judged according to the ROC curve. Why not recalculate the AUC in the validation data and select the corresponding threshold?

Answer:

Thank you very much for the question. The final step of the classifier construction is to define the prediction rule⁹. The classification rule in our penalized Cox regression model is simply a cutoff value. Proper validation requires validating the fully specified proposed prognostic model¹⁰. In our study, it includes the selected lncRNAs, their coefficients and the cutoff values for classification. Actually, researchers usually validate their models with the cutoff value developed from a training cohort¹¹⁻¹³, which are perfect and important methodological references for our work. Instead of recalculating the cutoff values in the validation cohorts, using the cutoff value from the training cohort could avoid overestimation of the prediction accuracy in the validation cohorts, verifying the reproducibility and robustness of the model. Therefore, we used the threshold developed in the training cohort to validate the lncRNA signature in the validation cohorts.

References:

9. Simon, R., Radmacher, M. D., Dobbin, K. & McShane, L. M. Pitfalls in the use of DNA microarray data for diagnostic and prognostic classification. *Journal of the National Cancer Institute* **95**, 14-18 (2003).
10. Altman, D. G., Vergouwe, Y., Royston, P. & Moons, K. G. Prognosis and prognostic research: validating a prognostic model. *BMJ* **338**, b605 (2009).
11. Xu, R. H. et al. Circulating tumour DNA methylation markers for diagnosis and prognosis of hepatocellular carcinoma. *Nat Mater* **16**, 1155-1161 (2017).
12. Scott, D. W. et al. Gene expression-based model using formalin-fixed paraffin-embedded biopsies predicts overall survival in advanced-stage classical Hodgkin lymphoma. *J Clin Oncol* **31**, 692-700 (2013).
13. Liu, N. et al. Prognostic value of a microRNA signature in nasopharyngeal carcinoma: a microRNA expression analysis. *The Lancet Oncology* **13**, 633-641 (2012).

4. In the result 'Biological characteristics related to the nine-lncRNA signature', the description of correlation analysis is relatively simple. Which method does the author use? This should be described.

Answer:

Thank you very much for the valuable comments. In this study, we investigated the functional contexts of the nine lncRNAs in our signature based on the guilt-by-association principle, which uses correlation information to predict new gene members in the functional call of genes¹⁴. This approach is based on correlation analysis between matching lncRNA and protein-coding mRNA expression in combination with enrichment strategies. It contains the following three steps: (i) for each lncRNA, an expression-correlation matrix (Spearman's rank) was built by combining lncRNA and matching mRNA expression data; (ii) the expression-correlation matrix was ranked by correlation coefficient in descending order; and (iii) the sorted expression-correlation matrix was used as input for gene set enrichment analysis (GSEA) (hallmark gene set list obtained from MSigDb and immune-related pathways from the ImmPort website¹⁵). Pathways with $P < 0.05$, $FDR < 0.25$ and absolute normalized enrichment score ($|NES| > 1$) were considered significant and were used to infer the functions of the corresponding lncRNAs.

In light of the reviewer's comments, we have added details about this functional enrichment analysis to the Methods section of the revised manuscript (Page 16, paragraph 2).

References:

14. Lefever, S. et al. DecodeRNA- predicting non-coding RNA functions using guilt-by-association.

Database (Oxford) 2017 (2017).

15. Bhattacharya, S. et al. ImmPort, toward repurposing of open access immunological assay data for translational and clinical research. *Sci Data* **5**, 180015 (2018).

Minor Point:

5. The AUC value related to the optimal cut-off value should be showed in the result.

Answer:

Thank you very much for the suggestion. We have added the AUC value related to the optimal cut-off value to the Results section of the revised manuscript (Page 7, paragraph 1).

REVIEWERS' COMMENTS

Reviewer #1 (Remarks to the Author):

The revised manuscript is greatly improved. The authors have addressed all my comments/concerns. I have no further comment.

Reviewer #2 (Remarks to the Author):

I thank the authors for their thoughtful and thorough responses to my comments. I am satisfied that all concerns have been addressed.

Reviewer #3 (Remarks to the Author):

The authors have addressed the majority of my concerns. However, the authors should compare their current method with those integrating lncRNAs and protein coding genes. Whether the integration will improve the predictive power?

Response to the reviewers' comments and questions:

Response to Reviewer #1:

The revised manuscript is greatly improved. The authors have addressed all my comments/concerns. I have no further comment.

Answer:

Thank you. We all appreciate your valuable comments, questions and suggestions, which have greatly helped us to improve our manuscript.

Response to Reviewer #2:

I thank the authors for their thoughtful and thorough responses to my comments. I am satisfied that all concerns have been addressed.

Answer:

Thank you very much for your valuable comments, questions and suggestions, which bring great improvement in our work.

Response to Reviewer #3:

The authors have addressed the majority of my concerns. However, the authors should compare their current method with those integrating lncRNAs and protein coding genes. Whether the integration will improve the predictive power?

Answer:

Thank you for your valuable questions. With limited data of mRNA available in our study, we tried to explore whether integrating mRNAs with lncRNAs could further enhance the prediction accuracy in the microarray data from LA-NPC samples in the discovery cohort (n = 38). We used ceRNA network to get candidate mRNAs¹⁻³. For each lncRNA in our signature, we used Spearman correlation test to get the top 5 mRNAs whose expression was positively correlated to lncRNAs. DIANA Tools (<https://diana.e-ce.uth.gr/lncbasev3/interactions>) was used to determine the interactions between lncRNAs and miRNAs, and TargetScan⁴ (Release 8.0, https://www.targetscan.org/vert_80/) was used to determine the interactions between mRNAs and miRNAs. If a pair of miRNA-mRNA interaction shared the same miRNA with a pair of

miRNA-lncRNA interaction, the mRNA was considered as the target of ceRNA network. In total, five mRNAs (*WDTC1*, *PDE7B*, *YPEL2*, *ITGA9*, *NXPE3*) were selected. Because the lncRNA signature developed with PCR data was not comparable to those integrating lncRNAs-mRNAs with microarray data, so we developed two models both with microarray data. One model incorporated the nine lncRNAs in our signature, and another incorporated the expression of the nine lncRNAs and five mRNAs, both using formula weighted by their coefficients in Cox analysis. ROC analysis discovered that the area under curve (AUC) of the lncRNA model was 0.93, and the AUC of the lncRNA-mRNA model was 0.97 (**Figure 1A**), which were not significantly different (Two-sided DeLong test, $P = 0.42$). In addition, we tried to a new method, random forest⁵, to develop models and similar result was observed. The AUC of the lncRNA model was 0.96, and the AUC of the lncRNA-mRNA model was 0.94, which were not significantly different (Two-sided DeLong test, $P = 0.71$) (**Figure 1B**). Based on these results, integrating data of lncRNAs and mRNAs may not further boost the efficacy, but more studies with larger sample size are needed before drawing a conclusion.

Figure 1. Performance of the lncRNA-mRNA model. ROC curve analysis of the lncRNA model and lncRNA-mRNA model for predicting distant metastasis in patients in the discovery cohort ($n = 38$) using (A) Cox analysis and (B) random forest. DeLong's test was used to estimate the P values.

References:

1. Cheng, Y. et al. Identification of circRNA-lncRNA-miRNA-mRNA Competitive Endogenous RNA Network as Novel Prognostic Markers for Acute Myeloid Leukemia. *Genes (Basel)* **11** (2020).

2. Wang, L. X. et al. Integrative Analysis of Long Noncoding RNA (lncRNA), microRNA (miRNA) and mRNA Expression and Construction of a Competing Endogenous RNA (ceRNA) Network in Metastatic Melanoma. *Med Sci Monit* **25**, 2896-2907 (2019).
3. Wang, W. et al. A novel mRNA-miRNA-lncRNA competing endogenous RNA triple sub-network associated with prognosis of pancreatic cancer. *Aging (Albany NY)* **11**, 2610-2627 (2019).
4. Lewis BP, Burge CB, Bartel DP. Conserved Seed Pairing, Often Flanked by Adenosines, Indicates that Thousands of Human Genes are MicroRNA Targets. *Cell*, **120**:15-20 (2005).
5. Breiman, L. Random forests. *Machine learning*, **45.1**, 5-32 (2001).